# Distribution and Morphology of α-Al, Si and Fe-Rich Phases in Al–Si–Fe Alloys under an Electromagnetic Field

**DOI:** 10.3390/ma16093304

**Published:** 2023-04-23

**Authors:** Piotr Mikolajczak

**Affiliations:** Institute of Materials Technology, Poznan University of Technology, Piotrowo 3, 60-965 Poznan, Poland; piotr.mikolajczak@put.poznan.pl; Tel.: +48-(61)-665-2804

**Keywords:** electromagnetic stirring, aluminum alloys, solidification, dendrites, rosettes, iron phases

## Abstract

Natural convection is present in all liquid alloys whereas forced convection may be applied as the method to improve material properties. To understand the effect of forced convection, the solidification in simple cylindrical samples was studied using a rotating magnetic field with a low cooling rate and low temperature gradient. The composition of Al–Si–Fe alloys was chosen to enable independent growth or joint growth of occurring α-Al, β-Al_5_FeSi, δ-AlFeSi_T4 phases and Si crystals and analysis of structure modifications. Stirring produced rosettes instead of equiaxed dendrites, which altered the secondary dendrite arm spacing and the specific surface of α-Al and also modified β-Al_5_FeSi. The melt flow caused a modification of iron rich δ-AlFeSi_T4 phases and gathered them inside the sample of the β/Si alloy, where δ together with Si were the first precipitating phases. The separation of δ and β phases and Si crystals was found by their joint growth along the monovariant line. A reduction in the amount of Si crystals and the formation of a thin Si-rich layer outside the sample was observed in the hypereutectic alloy. The separation and reduction in iron-rich phases may play a role in the removal of Fe from Al–Si alloys, and the control of Si may be applied in materials for the solar photovoltaic industry.

## 1. Introduction

From foils to advanced engineering applications, aluminum alloys have a broad range of uses [1] in the automotive, electrical and aerospace industries because of their unique combination of properties. As structural metals, technical aluminum alloys (e.g., 336.1, EN AB-44300 or EN AC-44300) are second only to steels. Among other things, the increasing demand for such materials is characterized by a high strength-to-weight ratio, ductility and high corrosion resistance and advanced manufacturing practices are required for greater control of microstructure, chemical composition and heat treatment [2,3].

During solidification, the following phenomena occur within the semisolid region: crystallization, solute redistribution, ripening, solid movement and inter-dendritic flow. Flemings [4] discovered metallic alloys with special rheological properties caused by the formation of a non-dendritic structure in the semisolid state, e.g., in aluminum alloys, which leads to a transformation of the dendritic phase into the α-Al phase, shaped as rosettes or spheroids (globular) [5]. Non-dendritic structures exhibit rheological properties [6] that improve the mechanical properties of composites [7] and alloys [8], thereby allowing for the production of advanced engineering parts [9] in rheocasting [10], thixowelding [11], semisolid metal processing (SSM) [12,13], thixoforming [14], through the use of cooling slopes [15], by magnetohydrodynamics (MHD) [16] or simply by mechanical stirring [17]. Melt flow induced by rotating magnetic fields (RMF) in a technology called electromagnetic stirring (EMS) [18,19], transforms the microstructure [20] and improves the properties of billets [21]. The development of melt conditioning (MC) [22] and high shear melt conditioning (HSMC) [23,24] seems to offer improvements in degassing, mechanical properties, defects and in situ microstructural control of second phase particles in aluminum alloys.

In cast aluminum alloys Cu, Si and Mg are the main alloying elements [1], whereas the main impurities that exist in recycled alloys are manganese (Mn), iron (Fe), copper (Cu), and zinc (Zn). The use of recycled foundry alloys containing even small amounts of iron (in Al max. 0.05%) results in the formation of Fe-rich intermetallics. β-Al_5_FeSi platelets and needles [25] have a detrimental effect on alloy properties [26], thus alloying elements such as Cr, Be, Mn and Mo have been used to modify the acicular β into the granular or skeleton morphology.

In terms of the β-Al_5_FeSi and δ-AlFeSi_T4 phases, the Al–Si–Fe system with its corresponding ternary phase diagram (Figure 1), the modification in morphology of phases present and the transformation of the whole microstructure under the influence of electromagnetic stirring are important. In this paper, the effect of stirring has been studied for solidifying alloys, in which the composition (Figure 1) was chosen in order to study the influence of melt flow on separated phases, precipitating individually (independent growth) or both together (joint growth), before the final eutectic reaction. According to the Al–Si–Fe ternary phase diagram, hypoeutectic, hypereutectic and eutectic alloys were determined.

Cylindrical samples were solidified with and without an induced rotating magnetic field (RMF) during the controlled slow cooling in insulated crucibles. Microstructures from optical metallography were measured using the ImageJ 1.51a software. The specific alloys with compositions located around the eutectic point (Figure 1) and property diagrams were calculated using thermodynamic software [27].

Two main motivations spurred the current study. Firstly, due to the continuous development of forced-flow manufacturing technologies [5,6,7,8,9,10,11,12,13,14,16,17,18,19,20,21,22,23,24], the modification of the microstructure was investigated on the whole sample and in detail on individual precipitates. Secondly, in terms of deleterious iron precipitates, the possibility of iron removal or its intermetallics separation was of interest.

## 2. Materials and Methods

The study encompassed ten aluminum alloys from the Al–Si–Fe system (Figure 1). These were selected in order to consider the influence of electromagnetic stirring on the phases precipitating alone (independently) or both together. In the first group of alloys, the precipitation of the first phase (independent growth) occurs as the only one, starting at a liquidus temperature and ending at a solidus temperature, whereas the remaining phases may form in the final eutectic reaction. The second group includes alloys where precipitation of the first two phases occurs simultaneously and together (joint growth) with the remaining phases forming at the solidus temperature. Essentially, the liquidus–solidus temperature range of 610–575 °C was chosen for six alloys, and for three additional alloys the range was narrower, i.e., 630–575 °C. The same liquidus temperature of 610 °C was chosen to maintain similar thermal conditions in the alloys with different compositions, while 630 °C should provide information on the effect of a narrower solidification range on the influence of fluid flow on the microstructure. The tenth alloy was chosen with a eutectic point composition with a very narrow solidification temperature range. In the first group of alloys (independent growth) were:-“α-Al-first” alloy (Figure 1, solid blue line)—the α-Al phase precipitates first from the liquid alloy starting at 610 °C (composition Al–Si7.837Fe0.521), and similarly, the “α-Al-2-first” alloy with the Al–Si4.861Fe0.306 composition where the α-Al phase also starts to precipitate first but at 630 °C. In both alloys, other phases (Al–Si eutectics and iron intermetallics) may precipitate at the eutectic point, at 575 °C;-“β-first” alloy (Figure 1, solid red line)—the β-Al_5_FeSi phase precipitates first from the liquid alloy starting at 610 °C (composition Al–Si12.795Fe1.705), and similarly, the “β-2-first” alloy with the Al–Si12.911Fe2.372 composition where the β-Al_5_FeSi phase also starts to precipitate first but at 630 °C. In both alloys the remaining phases (Al–Si eutectics and α-Al) precipitate at the eutectic point, at 575 °C;-“Si-first” alloy (Figure 1, solid green line) —Si crystals precipitate first from the liquid alloy starting at 610 °C (composition Al–Si14.877Fe0.871), and similarly, the “Si-2-first” alloy with the Al–Si16.187Fe0.858 composition where Si crystals also start to precipitate first but at 630 °C. In both alloys other phases (α-Al and β-Al_5_FeSi) precipitate at the eutectic point, at 575 °C.

In the second group, the precipitation of the first two phases occurs simultaneously (joint growth):-“α-Al/β” alloy (Figure 1, dashed violet line), α-Al and β-Al_5_FeSi phases precipitate simultaneously (along the monovariant line) first from the liquid alloy starting at 610 °C (composition Al–Si7.508Fe1.687), the remaining phases (Al–Si eutectics) precipitate at the eutectic point, at 575 °C;-“α-Al/Si” alloy (Figure 1, dashed yellow line), α-Al and Si crystals precipitate simultaneously (along the monovariant line, composition Al–Si12.587Fe0.443), other phases (Al–Si eutectics) precipitate at the eutectic point. Here, the liquidus temperature (not 610 °C) and composition were in the middle of the monovariant line, in order to keep iron and differ from the “eutectic point” alloy;-“β/Si” alloy (Figure 1, dashed grey line), δ phases, Si crystals (and β-Al_5_FeSi later) precipitate simultaneously first from the liquid alloy starting at 610 °C (composition Al–Si15.136Fe1.678), the remaining phases (Al–Si eutectics) precipitate at the eutectic point.

The tenth “eutectic point” alloy was selected with the composition Al–Si12.674Fe0.895 (Figure 1, brown point).

All the alloys were prepared from pure components: Fe (99.99+% HMW Hauner GmbH & Co. KG, Röttenbach, Germany), Si (99.9999% NewMet House, Essex, UK) and Al (99.999% HMW Hauner). The alloys were melted (min. 1.5 h), without the addition of a modifier, in a cylindrical graphite crucible (inner diameter of 38 mm) using an electric resistance furnace, and the entire melting process was conducted by flushing the crucible with argon and degassing the alloy.

The specimens (38 mm in diameter and 65 mm in height, Figure 2) were melted and solidified in the crucible, and after reaching a temperature of 800–805 °C, were transferred from the furnace into the solidification facility equipped with electric coils and Sibral insulation (Fiberfrax, Unifrax, Tonawanda, NY, USA). Cooling rates were determined by taking temperature measurements at the center of the specimen, as well as in the outer part and inside the crucible. The cooling rates were: R_800-liq_ = 0.518 (K/s), R_liq-sol_ = 0.89 (K/s), and R_sol-470_ = 0.232 (K/s) for the “α-Al-first” alloy without stirring, and for the fluid flow R_800-liq_ = 0.498 (K/s), R_liq-sol_ = 0.82 (K/s) and R_sol-470_ = 0.257 (K/s). The temperature gradients between the specimen’s center and the outer part of the specimen were: without stirring G_800-liq_ = 0.189 (K/mm) and G_liq-470_ = 0.127 (K/mm), and with flow G_800-liq_ = 0.178 (K/mm) and G_liq-470_ = 0.116 (K/mm). The low temperature gradient and low cooling rate showed that the cooling of the crucible and the specimen together ensured slow simultaneous solidification within the entire specimen and the formation of equiaxed dendrites (without stirring RMF 0 mT).

Electromagnetic coils powered from an autotransformer at 10 A and 45 V at a frequency of 50 Hz generated a rotating magnetic field (RMF) with a flux density of 11 mT (Magnetic Field Meter AC/DC, MF100, Extech Instruments, Nashua, New Hampshire, NH, USA). Regarding the rotating sample cylinder, the estimated rotational speed of the alloy reached 2.1 s^−1^.

The solidified samples (Figure 2) were cut (Mecatome T255/300, Presi, Grenoble, France) at a height of 10 mm from the bottom for the transverse cross-section. The microsections were prepared using a standard metallographic facility (Mecatech 250 SPC, Presi, Grenoble, France), and observed with a light optical microscope LOM (Nikon Eclipse MA200, Tokyo, Japan) and imaging software (NIS Elements 5.21.03, Nikon, Tokyo, Japan). In total, 20 cross-sections from 20 experiments were analyzed (10 alloys, each with and without a forced flow). The microstructure was analyzed using the ImageJ 1.51a software (called also alternatively Fiji, National Institutes of Health, Bethesda, MD, USA) on the cross-section in four specific areas (the white-filled rectangles in Figure 2), in one larger area (white dashed line rectangle) and in arched areas (yellow dashed line) using the image stitching technique and magnifications of 50×, 100×, 200× and 500×.

The following parameters were determined on the cross-section: average length L_β_ and number density n_β_ of β-Al_5_FeSi platelets, average length L_δ_ and number density n_δ_ of iron δ-phases, average length L_Si_ and number density n_Si_ of Si crystal, specific surface of dendrites S_v_, secondary dendrite arm spacing λ_2_, and eutectic spacing λ_E_ for Al–Si eutectics. In the measurement of 21,389 β-Al_5_FeSi intermetallics, only needles with a length/thickness ratio > 5 and thickness > 3 µm were considered. In the measurement of 1491 iron δ-phases, all needles were considered. The specific surface of dendrites S_v_ was measured from the enclosed area and the perimeter of α-Al. The secondary dendrite arm spacing λ_2_ was calculated by averaging the distance between 10 and 50 adjacent side branches. For Al–Si eutectics, the spacing λ_E_ was measured by averaging the distance between adjacent plates.

In the present paper, the author studied precipitations, such as Al–Si eutectics, dendritic α-Al, needle or platelet shaped β-Al_5_FeSi and δ-AlFeSi_T4 phases; these are all well-known from many other studies (e.g., in [1,2,3,4,5,6,25,26,28,29,30,31,32,33,34]) concerning aluminum alloy phases. The precipitation sequence was first considered theoretically according to the Al–Si–Fe phase diagram, but the exact precipitation of only one first phase (e.g., α-Al in the “α-Al-first” alloy) or both (e.g., dendritic α-Al and β-Al_5_FeSi in “α-Al/β” alloy) required a precise determination of the alloy composition, which was calculated using the Thermo-Calc software [27]. Widely used by materials scientists and engineers, the software was used to determine property diagrams, the sequence of precipitation of phases, the ternary phase diagram and the Scheil solidification calculations.

## 3. Results

The results include figures presenting microsections of the alloys, tables with measured and counted parameters characterizing precipitated phases, and a table and property diagrams describing the solidification sequence and mass fraction of the forming phases.

### 3.1. Microstructure

Micrographs (Figure 3) of the typical structure obtained in experiments for solidification, with and without forced convection, for the α-Al-first alloy (Al–Si7.837Fe0.521), show clearly formed characteristic α-Al equiaxed dendrites, whereas for electromagnetic stirring RMF, α-Al formed as rosettes and as spheroids (globular forms), or as not fully formed minor dendrites. Figure 4 shows similar microstructures for the α-Al-2-first alloy (Al–Si4.861Fe0.306, T_L_ = 630 °C), with more α-Al phase and less eutectics, but with the same transition from dendritic α-Al to rosettes/spheroids structure.

Figure 5 presents the microstructure of a β-first alloy (Al–Si12.795Fe1.705) specimen solidified with and without melt flow, with clearly visible β-Al_5_FeSi phases evenly distributed on the specimen. Higher magnification (Figure 6) shows β-Al_5_FeSi needles located in Al–Si eutectics with some small α-Al precipitations occurring shaped as dendrites and rosettes. A similar microstructure (Figure 7a,b) can be observed for the β-2-first alloy (Al–Si12.911Fe2.372) with a higher iron content and liquidus temperature of 630 °C. For both alloys, when solidifying without stirring (Figure 5a and Figure 7a), it is possible to observe the presence of very long β needles reaching up to even several millimeters as well as very small β needles, whereas by melt flow (Figure 5b and Figure 7b), the lengths seem to be more equal for the occurring needles.

The microstructure of the Si-first alloy (Al–Si14.877Fe0.871) is shown in Figure 7c,d, and in detail in Figure 8. In the case of solidification without stirring, it is possible to see the internal area A reaching a dimension of 0.5 of radius R (0.5R), filled almost exclusively with Al–Si eutectics, whereas the external area B has many Si crystals. In the case of solidification with stirring (Figure 7d), a thin layer rich in Si was formed, and the outside area B is smaller, starting from 0.8R to the edge of the specimen, and the amount and dimension of Si crystals seem to be smaller. In the Si-2-first alloy (Al–Si16.187Fe0.858), because of the higher amount of silicon, the occurrence of the two areas is similar (Figure 9), and the melt stirring also changes the dimensions of the areas, from radius 0.7R to 0.85R. In the Si-first and Si-2-first alloys, fluid flow seems to decrease the amount and location of Si crystals, forming a thin layer. In addition to Si crystals and eutectics, some β-Al_5_FeSi phases occurred in all areas of both alloys.

For the α-Al/β alloy (Al–Si7.508Fe1.687) the microstructure across the specimen and at the solidification, without (Figure 10a) and with stirring (Figure 10b), shows a uniform distribution of α-Al and β-Al_5_FeSi phases and eutectics present. For the melt flow, both well-developed dendrites and rosettes are present (Figure 11a,b), and it is characteristic that many α-Al phases are shaped like a “bunch of grapes”.

The microstructure of the α-Al/Si alloy (Al–Si12.587Fe0.443) is almost completely filled with Al–Si eutectics, with some α-Al and β-Al_5_FeSi phases. In the case of solidification without stirring, several large dendrites of up to 5 mm in length grew (Figure 12a), while the forced flow resulted in much smaller α-Al shaped rosettes (Figure 12b).

On the microstructure of the β/Si alloy (Al–Si15.136Fe1.678), there appeared Si crystals, β-Al_5_FeSi and δ-phases and Al–Si eutectics. They were more or less evenly distributed across the specimen in the case of solidification without stirring (Figure 13a), whereas in the case of induced melt flow, three characteristic areas occurred. The internal area A with a radius of up to 0.63R is filled mainly with δ-phases and Al–Si eutectics (Figure 13b, Figure 14a and Figure 15), whereas area B contains mainly β-Al_5_FeSi in Al–Si eutectics (Figure 14b and Figure 15). In the third area C, from a radius of 0.9R to the edge of the specimen, there were mainly Si crystals in Al–Si eutectics (Figure 13b and Figure 15). The specimen of the β/Si alloy has clearly recognizable various structures, where β, δ phases and Si crystals were distinguished in the three areas formed.

The microstructure across the specimen of eutectic point alloy Al–Si12.674Fe0.895, for solidification with and without stirring presents a uniform distribution of phases present, mainly Al–Si eutectics.

### 3.2. Parameters Characterising the Microstructure

The modification of the microstructure by electromagnetic stirring is characterized by the parameters counted and measured in specified areas (Figure 2, Figure 7c,d, Figure 9a,b and Figure 13a,b) on the cross-section. Such a methodology provided reliable results presented in Table 1, Table 2 and Table 3, across solidified samples, along with a proper overview of all the specimens. The measurements carried out on a large number of grains, crystals and intermetallics ensured the high quality of the results. The standard deviation is given in square brackets (Table 1, Table 2 and Table 3), and the number of inspected grains and dendrite arms, e.g., for the whole α-Al-first alloy specimen solidified with stirring is shown in parentheses. The secondary dendrite arm spacing λ_2_ is 89 μm (Table 1), with a standard deviation of σ = 7.5 μm and was measured on 37 grains and 222 secondary arms. The forced flow changed λ_2_ by −6% in comparison to the solidification without stirring. The secondary spacing is 92 μm (standard deviation σ = 7.0 μm) in area A (Figure 2), 94 μm (σ = 9.4 μm) in area B, 87 μm (σ = 5.6 μm) in area C and 84 μm (σ = 6.1 μm) in area D (Table 1). The data for other parameters were similar. For the Si-first alloy, the specific surface S_v_ is 0.087 μm^−1^ (σ = 0.008 μm^−1^) and is equal to the value for area A, because no dendrites were found in areas B, C and D. A dash in Table 1, Table 2 and Table 3 “-“ indicates a lack of dendrites or intermetallics in the selected area. Due to the observed areas A, B and C (Figure 7c,d, Figure 9 and Figure 13) with differing amounts of Si crystals and intermetallics, results were presented for only the two or three areas shown. For the Si-first alloy, the average length of β-Al_5_FeSi phases is 394 μm (Table 2) over the whole cross-section, whereas these were 326 μm in area A and 432 μm in area B, without determination for areas C and D (Figure 7c,d). A similar methodology was adopted for the Si-2-first (Figure 9) and β/Si (Figure 13) alloys, and the measurement of the Al–Si eutectic spacing, δ-phases and Si crystals (Table 2 and Table 3).

For the α-Al-first and α-Al-2-first alloys, the induced fluid flow caused an approximately 6% and 9% decrease in secondary dendrite arm spacing λ_2_, from 95 µm to 89 µm and from 112 µm to 102 µm, respectively. Throughout the specimen, in areas from A to D, λ_2_ seems to have similar values, when considering the standard deviation. A 25% reduction in λ_2_ values was also observed in the eutectic point alloy. For all other alloys, the dendrite spacing increased under melt flow, e.g., 60% for the β-first alloy, 170% for the β-2-first alloy, 255% for the β/Si alloy and 11% for the α-Al/β alloy. The reductions in specific surface S_v_ under stirring for the α-Al-first and α-Al-2-first alloys were 41% and 32%, respectively, well above the standard deviation range, and with similar values throughout the specimen, in areas A–D. Only for the eutectic point alloy was S_v_ increased by 25%. In the other alloys the following reduction in S_v_ was observed: 42% for the β-first alloy, 57% for the β-2-first alloy, 255% for the β/Si alloy, 53% for the β/Si alloy and only 3% for the α-Al/β alloy.

The average length L_β_ of β phases (Table 2) decreased by 14% under flow for the α-Al-first alloys, whereas it remained unchanged (+2%) for the α-Al-2-first alloy (σ = 0.224 μm and 0.176 μm). For the β-first and β-2-first alloys, L_β_ increased by 92% and 76%, respectively, and by 17% for the α-Al/β alloy. The histogram (Figure 16) shows the number of measured β-Al_5_FeSi needles, grouped into sets of lengths, e.g., 100–200 µm, for the β-first alloy, and shows that there are more phases for solidification without stirring, with lengths ranging between 0–600 µm, and also with lengths exceeding 2000 µm. For the other alloys, the Si-first, α-Al/Si, β/Si and eutectic point alloys, the length L_β_ was reduced in the range from 10% to 31% and was unchanged for the Si-2-first alloy. For the α-Al-first and α-Al-2-first alloys, the induced fluid flow caused an increase (130% and 33%) in the number density n_β_ from 6.55 to 15.04 mm^−2^ and from 6.79 to 9.02 mm^−2^, respectively, whereas for the β-first and β-2-first alloys the values decreased 71% and 70%. Moreover, across the specimens, from the center to the outer part (from area A to D), the number density n_β_ changed considerably. An increase in the number density n_β_ was also noted for the Si-first, Si-2-first, α-Al/Si and eutectic point alloys, whereas for α-Al/β and β/Si, it was decreased. The convergence of the results for the L_β_ and n_β_ values and their modification values, in the β-first and β-2-first alloys, authenticates the methodology of the experiments, measurements and results in this study.

The effect of electromagnetic stirring on (α-Al)-Si eutectic (Table 2) is unclear; both decreases and increases were observed for individual alloys and there is no direct modification in the eutectic spacing λ_E_. For the β-first alloy, the eutectic spacing λ_E_ decreased by 6% from 7.10 to 6.68 µm, with standard deviation values of 0.77 and 0.74 µm. Meanwhile, for the β-2-first alloy, λ_E_ increased by 27% from 6.87 to 8.75 µm, with standard deviation values of 1.59 and 1.56 µm.

The average length L_δ_ of δ phases (Table 3) increased by 28% under flow for the β/Si alloy, from 240 to 307 µm, and also changed considerably across the specimen, while the number density n_δ_ decreased by 58%. Noteworthy are the changes in the number density n_δ_ across the specimen and the δ phases concentrated in the specimen center (n_δ_ = 5.60 mm^−2^), whereas in the outer areas B and C (Figure 13) they are almost completely absent (n_δ_ = 0.09 and 0.70 mm^−2^).

For Si-crystals (Table 3), the average length L_Si_ increased by 17% and 20%, respectively, in the outer areas B (Figure 7c,d and Figure 9) in the Si-first and Si-2-first alloys, while the number density n_Si_ decreased considerably by 89% and 74%. For the β/Si alloy, L_Si_ increased by 51% from 565 to 851 µm, while n_Si_ decreased by 19%. A significant decrease in the number density was noticed in area B where the density decreased significantly from 0.248 to 0.060 mm^−2^.

### 3.3. Precipitation Sequence

In order to meet the assumptions defined in the methodology, precise alloy compositions were calculated using Thermo-Calc [27] software. Similarly, thermodynamic calculations determined the precipitation sequence and mass fraction of the phases as presented in Table 4 and in Figure 17 and Figure 18, also for the selected three alloys described in detail below.

On the basis of the ternary Al–Si–Fe phase diagram (Figure 1), thermodynamic calculations [27] and property diagram (Figure 17a), for the case where *α*-Al is the first growing phase, the alloy with composition Al7.837Fe0.521 was designated. For what is referred to as the *α*-Al-first alloy (Table 4) in this study: first *α*-Al forms at 610 °C (L → *α*-Al + L), then the liquid is enriched in Si and Fe to a concentration of 12.565%Si and 0.885%Fe until the eutectic reaction is reached at 575.83 °C. Then, the eutectic groove L → L + *α*-Al + Si starts at 575.83 °C and ends at 575.02 °C with the composition of the rest of the liquid being 12.674%Si and 0.894%Fe; this is followed by the final eutectic reaction L →*α*-Al + β-Al_9_Fe_2_Si_2_ + Si. At this point (temperature 575.02 °C), the mass fraction of *α*-Al reaches f*_α_*_-Al_ = 91.96%, the mass fraction of β-Al_9_Fe_2_Si_2_ reaches f_β-Al5FeSi_ = 1.92%, and the eutectics mass fraction reaches f_Eut_ = 6.12%. Detailed thermodynamic calculations and their precision showed the occurrence of the L → L + *α*-Al + Si reaction, which was unintentional; it happened only in a very short temperature range, and the difference in concentrations of, e.g., Si was between 12.565% and 12.674%. From a practical point of view, this is not relevant or even visible in the phase diagram (Figure 1). Practically and intentionally, we only have two reactions; L → *α*-Al + L for the temperature range 610–575.83 °C and L → *α*-Al + β-Al_9_Fe_2_Si_2_ + Si for the range 575.83–575.02 °C.

For the alloy in the Al–Si–Fe system, where β-Al_5_FeSi is the first growing phase, the alloy with composition Al–Si12.795Fe1.705 was designated. For what is referred to as the β-first alloy in this study (Table 4, Figure 1 and Figure 17b): first β-Al_5_FeSi is formed at 610 °C (L → β-Al_5_FeSi + L), then the liquid is enriched in Si and Fe to a concentration of 12.675%Si and 0.910%Fe until the eutectic reaction is reached at 575.21 °C. Then, the eutectic groove L → L + β-Al_5_FeSi + Si starts at 575.21 °C and ends at 575.02 °C, with the composition of the rest of the liquid being 12.674%Si and 0.894%Fe; this is followed by the final eutectic reaction L → *α*-Al + β-Al_5_FeSi + Si. At this point (temperature 575.02 °C), the mass fraction of *α*-Al reaches f*_α_*_-Al_ = 83.17%, the mass fraction of β-Al_9_Fe_2_Si_2_ reaches f_β-Al5FeSi_ = 6.33% and the mass fraction of eutectics reaches f_Eut_ = 10.50%. Detailed thermodynamic calculations and their precision showed the occurrence of the L → L + β-Al_5_FeSi + Si reaction, which was unintentional, occurring only over a very large temperature range, the difference in concentrations of e.g., Si were between 12.675% and 12.674%, and Fe between 0.910% and 0.894%. From a practical point of view, this is not relevant and not even visible in the phase diagram (Figure 1). For this alloy, practically and intentionally, we only have two reactions L → β-Al_9_Fe_2_Si_2_ + L for the temperature range 610–575.21 °C, and L → *α*-Al + β-Al_5_FeSi + Si for the temperature range 575.21–575.02 °C.

The alloy with composition Al–Si7.508Fe1.687 was designated, in which there is a common parallel (joint) growth of *α*-Al and β-Al_5_FeSi phases. For what is referred to as the *α*-Al/β alloy in this study (Table 4, Figure 1 and Figure 18): both *α*-Al and β-Al_5_FeSi start to form at 610 °C (L → *α*-Al + β-Al_5_FeSi + L), then the liquid, according to the eutectic groove, is enriched in Si and Fe to concentrations of 12.563%Si and 0.906%Fe until the eutectic reaction is reached at 575.03 °C. Then, at this point, the final eutectic reaction L → *α*-Al + β-Al_5_FeSi + Si occurs (temperature 575.02 °C). In the solidified alloy, at 575.02 °C, the mass fraction of *α*-Al reaches f*_α_*_-Al_ = 88.60%, the mass fraction of β-Al_5_FeSi reaches f_β-Al5FeSi_ = 6.26%, and the mass fraction of eutectics reaches f_Eut_ = 5.14%. For this alloy, practically and intentionally, we have two reactions L → *α*-Al + β-Al_5_FeSi + L for the temperature range 610–575.03 °C, and L → *α*-Al + β-Al_5_FeSi + Si for the temperature range 575.03–575.02 °C.

## 4. Discussion

The changes in microstructure were caused by a forced flow, which include the transformation from dendritic structures to rosettes or spheroids, as well as changes in the dimensions, amounts, and distributions of intermetallics, Si crystals, and Al-Si eutectics all require discussion. Here, the measured parameters and the various effects of stirring on the growing phases in the studied alloys will be discussed and analyzed.

### 4.1. Spheroids, Rosettes and Dendrites

In the studied alloys, α-Al crystals formed as fully shaped dendrites, rosettes and some globular grains. The rosettes appear to be the ripened arms of deformed dendritic crystals and build up as a result of the rotation of the dendrite tip during growth.

In numerical simulations, Das et al. [35] demonstrated that a tendency for dendritic growth may be diminished by turbulence and high shear rate, and globularization of the particle occurs as a result of the removal of constitutional undercooling at the solid–liquid interface and rotation of the primary solid phase. Birol [36] stated that the globular phase produced was due to the forced flow which caused the uniform chemical composition near the solid–liquid interface. Li et al. in experiments on succinonitrile (SCN)-5% water [37] supported the idea that the non-dendritic microstructure results from globular growth and natural nucleation. Martinez and Flemings [38] proved that intensive stirring leads to spheroidal forms just below the liquidus temperature in aluminum alloy. Ji et al. [39] obtained a spherical morphology instead of a dendrite or rosette in high shear rate solidification. The formation of spheroids requires a very high shear rate produced by intense stirring. The spheroidal forms observed in the present study appear to be a part of the dendrites as the rotational speed of melt was estimated to be as low as 2.1 s^−1^.

Based on the solidification of the AlCu10 alloy by forced convection [40], the authors suggested that the rosette-shaped clusters constitute the ripened arms of deformed dendritic crystals. In experiments with liquid alloys moving along a cooled plate, Birol [41] demonstrated that crystals have the ability to grow as agglomerates as a result of collisions with each other, building rosettes, fully shaped dendrites, and some globular grains. From cellular automaton simulations, Mullis [42] found that melt flow induces rotation of the dendrite tip leading to rosette precipitation without external mechanical interaction.

For equiaxed grain morphologies formed by equiaxed solidification, the microstructure is characterized by grain size, secondary dendrite arm spacing λ_2_, distance between grains, number of grains [43] and specific surface S_v_ [44], whereas in directional solidification, it is characterized by primary λ_1_ and secondary λ_2_ = λ_SDAS_ dendrite arm spacing [43,45,46,47].

Secondary dendrite arms start to form as perturbations located close to the dendrite tip. They then grow as cell-like structures and eventually form independent arms located in parallel to each other, where coarsening may determine the distance between the secondary arms λ_2_ [48,49]. Given on the idea that dendrite coarsening is diffusion controlled, this has led to the formulation of mathematical models [46,50,51,52], where secondary dendrite arm spacing λ_2_ = λ_SDAS_ is a function of the local solidification time t:(1)$$\lambda_{2}=c_{1}\cdot t^{n1}$$
where n_1_ = 0.48 for convective mass transport and 0.33 for the diffusive regime [53] and c_1_ materials coefficient [46,50,52,54,55,56]. Moreover, many numerical models for simulations [47,57] have been developed.

Mullis [58] found that flow from the tip of the secondary arm towards the root would extend the ripening rate, whereas enhanced ripening would be caused by flux in the opposite direction. Due to the increase in flow caused by coarsening, Diepers et al. [59] proposed a modification of the exponent n_1_ from 0.33 for diffusive ripening to 0.5. The results agree with experiments by Steinbach [53], Kasperovich [60], Ratke and Thieringer [61] on the convective ripening theory. According to the value of the c_1_ coefficient from [53] applied in the power law expressions of solidification time, Steinbach [53] proposed an exponent value n_1_ (in Equation (1)) amounting to 0.48 by flow caused by magnetic stirring (6 mT), instead of 0.36 in a solute-controlled (without stirring) system.

For the α-Al-first and α-Al-2-first alloys (Table 1), the induced fluid flow caused an approximate 6% and 9% decrease in the secondary dendrite arm spacing λ_2_, from 95 µm to 89 µm and from 112 µm to 102 µm, respectively. Throughout the specimen, in areas from A to D, λ_2_ seem to have similar values when considering the standard deviation σ. A 25% reduction in λ_2_ values was also observed in the eutectic point alloy. For all other alloys, the dendrite spacing increased under melt flow, e.g., by 60% for the β-first alloy, by 170% for the β-2-first alloy, by 255% for the β/Si alloy and by 11% for the α-Al/β alloy. In the α-Al-first, α-Al-2-first and α-Al/β alloys, the α-Al is dominant, whereas in the other alloys, its amount is small and almost negligible.

Stirring increased the solidification time (Table 1) for the α-Al-first alloy and this, according to Equation (1), for λ_2_, should mean an increase in the secondary spacing λ_2_, but in our case, it actually decreased. For the alloy solidified without stirring λ_2_ = 95 µm, but with flow, based on the measured solidification time, the calculated secondary spacing (λ_2_ = 100 µm) is still larger than the measured one (λ_2_ = 89 µm). In order to reach the measured value of 100 µm, the exponent n_1_ in equation (1) should be lower (n_1_ = 0.311), even lower than n_1_ for diffusive ripening (n_1_ = 0.33 without stirring) [53]. This means a decrease from 0.330 to 0.311, which is contrary to the literature data [53,59], suggesting an increase in n_1_ caused by convective ripening. This is similar for the α-Al-2-first alloy, where n_1_ decreased from 0.330 to 0.317.

For the β-first, β-2-first, Si-first and Si-2-first alloys, according to Equation (1), the solidification time and the measured secondary spacing λ_2_, in order to achieve calculated λ_2_ values equal to the measured ones, the exponent n_1_ by stirring should be higher, increasing from 0.33 for diffusive mass transport, to values such as 0.41, 0.49, 0.515 and 0.38, respectively, for convective ripening, and this is partially convergent with the values of 0.47–0.50 found in the literature [53,59] for directional solidification. For the α-Al/β, α-Al/Si and β/Si alloys and the eutectic point alloy, n_1_ changed from 0.33 to 0.345, 0.383, 0.53 and 0.28, respectively. The behavior of the exponents may suggest a lack of melt flow by electromagnetic stirring for the α-Al-first and α-Al-2-first alloys and the eutectic point alloy, or a significant reduction in the flow during the coarsening of the secondary arms. Regarding rosettes and the rarely occurring dendrites in electromagnetically stirred specimens, the presence of dendrites implies that the flow was very low in these small areas, and it is likely that secondary spacing formed more under diffusive than convective conditions. For the β-first, β-2-first, Si-first and Si-2-first alloys, where the mass fraction of β phases and Si crystals is smaller than that of α-Al in the α-Al-first and α-Al-2-first alloys, as seen in the microstructures (Figure 3, Figure 4, Figure 6, Figure 8 and Figure 11), property diagrams (Figure 17 and Figure 18) and Table 1, Table 2 and Table 3, there appear to be better conditions for an intensive and long-lasting forced flow.

The results for λ_2_ and n_1_ in the α-Al-first and α-Al-2-first alloys, as well as the eutectic point alloy, are in contrast to [53,59], where λ_2_ decreases by only 6% and 9%. This discrepancy supports the idea that there is insufficient convection, which affects the shape of dendrites and limits diffusive conditions to a local scale. In the case of other alloys, where λ_2_ undergoes significant changes and n_1_ increases, there appears to be no slowing down of the flow.

Marsh and Glicksman [62] found that, the specific interfacial area S_v_ was proportional to the solidification time t and proposed the following relationship:(2)$$S_{V}\;\sim \;t^{-1/3}$$

In the hold for 200 min directional solidification of an AlCu30 alloy, Kasperovich and Genau [63] found a decrease in S_v_ from 0.077 to 0.035 µm^−1^ with increasing electromagnetic stirring, and generally for holding times from 20 to 500 min, S_v_ showed values in the range 0.04–0.22 µm^−1^.

The specific surface S_v_ decreased (Table 1) under stirring for the α-Al-first and α-Al-2-first alloys by 41% and 32%, respectively, much more than the standard deviation range, and by similar values throughout the specimen, in areas A–D. Only for the eutectic point alloy was S_v_ increased by 25%. In the other alloys a reduction in S_v_ was observed: 42% for the β-first alloy, 57% for the β-2-first alloy, 255% for the β/Si alloy, 53% for the β/Si alloy, and only 3% for the α-Al/β alloy. For the calculation of S_v_, based on the measured solidification time and (2), a coefficient describing proportionality should be applied. The values of the coefficient for the alloys studied here range between 86 and 411 for solidification without forced convection, and between 99 and 600 for stirring. S_v_ changes more significantly than λ_2_ and seems to be a more effective indicator signaling the presence of melt flow.

The specific surface S_v_ decreased under fluid flow for almost all the alloys studied here, indicating that α-Al precipitates are more oval under stirring and that they are also larger structures or shapes. According to the above analysis, the best convergence of solidification time, dendrite arm spacing λ_2_ and specific surface S_v_ was found for the α-Al/β alloy. Common growth of α-Al phases and β needles seems to block the forced flow as the amount of solid phases increases during solidification.

### 4.2. Eutectics

In eutectics [43,45], the growth mechanism includes the cooperative growth of two (or more) phases by diffusion, and also separated growth without any exchange of solute between phases [46]. According to Jackson and Hunt [45,64], the eutectic spacing depends primarily on the solidification front velocity and materials coefficient, which was confirmed by Steinbach [53] in the directional solidification of the Al–Si7Mg0.6 alloy, where a reduction in the eutectic spacing with higher solidification (ranging between 10 and 120 μm/s) was observed. The results for Al–Si alloys reported by Sous [65] and Al–Si–Fe alloys by Mikolajczak [28] in experiments similar to [53] did not show any significant coincidence of the eutectic spacing with flow at 3 mT and 6 mT. The tendency to collect eutectics in the sample center as discovered by Ren and Junze may play a certain role [66]. In the Al-Si5Fe1.0 alloy, the eutectic reaction takes place at 575°C [29]. When the directional solidification is carried out with a temperature gradient of 3 K/mm, the length of the mushy zone is around 18 mm, whereas the eutectic zone is only about 1-3 mm. This suggests that convection may be limited. In the equiaxed solidification of AlCuSi alloys [67], at a temperature gradient of about 0.141 K/mm and a cooling rate of 0.108 K/s, and similarly in AlMgSi alloys [30], the fluid flow seems not to be diminished by the presence of equiaxed dendrites moving freely in the liquid with growing eutectic phases. However, in Mg-containing alloys [30] close to the solidus temperature, the fluid flow was found to be reduced by Mg_2_Si phases.

Table 2 shows that for the α-Al-first alloy forced convection increases the eutectic spacing λ_E_ by about 20%—from 6.5 µm to 7.8 µm—with a standard deviation of σ = 0.43 and 0.81 µm, these changes appear small. For the very similar α-Al-2-first alloy, the forced flow increases λ_E_ more strongly, by about 98% (from 2.99 to 5.91 µm) with a standard deviation of σ = 0.27 and 0.45 µm. In both alloys, the amount of eutectics in comparison to α-Al is small (Table 4), eutectics form, with well-developed and dense dendritic or rosette structures where the flow should be reduced. In similar alloys, β-first and β-2-first, the changes in λ_E_ are −6% and +27%, taking into account the standard deviation, both are negligible. The same is true for the Si-first and Si-2-first alloys. For the α-Al/β alloy, in which the amount of eutectics is small in comparison to the α-Al and β phases (Table 4), λ_E_ increased by 105% from 4.44 µm to 9.11 µm, by σ = 0.40 and 1.78 µm, respectively. However, with the excess of eutectics in the eutectic point alloy (Table 4), λ_E_ stayed unchanged, 14.66 µm and 15.02 µm (Table 2) and σ = 1.54 and 1.89 µm.

The unclear effect of flow on Al–Si eutectics during bulk solidification requires further investigation, including the use of a stronger electromagnetic field or mechanical stirring, smaller specimen sizes and the morphology of eutectic cells concentrated on eutectic-rich alloys.

### 4.3. β-Al_5_FeSi Intermetallics

The use of electromagnetic stirring by Fang et al. [32] resulted in the complete elimination of β-Al_5_FeSi phases with lengths of 95–110 µm in the Al–Si8Cu3Fe1.3 alloy, and a reduction in the average length from 75 µm to 15 µm for the Al–Si7Mg0.2–0.6Fe0.5 alloy. Nafisi et al. [31] found a shortening of Fe-rich intermetallic phases under electromagnetically induced force flow. For specimens solidified in a copper mold, the length decreased in the range of L_β_ = 4.5–5 µm, and from L_β_ = 9–10 µm to 7–8 µm in a sand mold. Steinbach et al. [33] in the Al–Si7Fe1.0 alloy specimens, with a diameter of 8 mm, solidified directionally by the rotating magnetic field, observed the growth of about 280 µm long β platelets in the eutectic microstructure, whereas without a forced flow, the β-Al_5_FeSi reached a length of 160 µm. In the directional solidification of Al–Si–Fe alloys, Mikolajczak and Ratke [34] presented a 9% increase in the length of β needles in the eutectic-rich microstructure under stirring, while there was a 20% shortening in the dendritic microstructure. A similar shortening was found for equiaxed solidification in the Al–Si5Fe1 alloy [59], whereas the addition of Mg weakened this effect as a consequence of a decreased flow in the presence of Mg_2_Si phases. For AlCuSi alloys [67], a similar methodology leading to equiaxed microstructure was used to decrease or increase the length of β needles, depending on the detailed composition and precipitation sequence of phases. Analysis of histograms [63] demonstrated that the decrease in the average length L_β_ of all β-Al_5_FeSi precipitates studied was due to an increase in the number density of the small, 5–40 µm long phases.

The number density [31] increased under a forced flow only slightly for the Al–Si6.8Fe0.8 alloy solidified in the copper mold, whereas in the sand mold, there was an increase from n_β_ = 600–1200 mm^−2^ to n_β_ = 800–2600 mm^−2^. In the directional solidification, Mikolajczak and Ratke [34] showed an increase in the number density n_β_ in the eutectic microstructure (42%) and in the outer dendritic area (17%) of cylindrical specimens. In the equiaxed solidification of the Al–Si5Fe alloy [30], the number density increased by 47%; however, with the addition of Mg, the changes were smaller depending on the composition of the alloys studied. For AlCuSi alloys [67], decreases or increases in the number density depending on the composition of the alloys studied were recorded using a similar methodology.

In the alloys in which α-Al precipitates as the first phase (Table 4, Figure 17a), the average length L_β_ of the β phases (Table 2) decreased under flow by 14% for the α-Al-first alloy, whereas it remained almost unchanged (+2%) for the α-Al-2-first alloy (with standard deviations of σ = 0.224 μm and 0.176 μm). The currently observed β phase shortening is consistent with the results obtained in [30,31,32,34], where alloys, in which α-Al phase precipitated first, were studied. In a study similar to the present one, a −20% change in length was observed in the Al–Si5Fe1 alloy [30] and −30% in the AlCu4Si6Fe1 alloy [67] during bulk solidification with a similar chemical composition, where the α-Al phase also grew first. For the α-Al-2-first alloy, it seems that the limit value of α-Al phase mass fraction was reached; there is too much α-Al, and the inter-dendritic cavities are insufficient for a positive β phase shortening effect. For the β-first and β-2-first alloys, L_β_ increased by 92% and 76%, respectively, and by 17% for the α-Al/β alloy, where β increased as the only solid phase from liquidus to solidus temperature (Table 4, Figure 17b). The increase in the β length is close to the observations made by Steinbach [33] and Mikolajczak and Ratke [34]. The results are similar to the AlCu10Si10Fe1 and AlCu4Si6Fe2 alloys [67], where forced convection resulted in an increase in the average length of β by 23% and 76%, respectively. For the AlCu10Si10Fe1.0 and AlCu4Si6Fe2 alloys, the β phases start to grow first, with about 50% and 30% of the Fe-rich phases precipitating before the α-Al phase. In the currently studied alloys, according to calculations (Table 4), 100% of β grew before α-Al and Al–Si eutectics started to precipitate. Comparing the changes of 92% for the β-first alloy, 76% for the β-2-first alloy and 76% for the AlCu4Si6Fe2 alloy [67], the iron content is 1.705%, 2.372% and 2.00% respectively, so the liquidus temperature and the first stages of solidification determine the potential for the increase in the β length. The histogram (Figure 16) shows the number of measured β-Al_5_FeSi needles, grouped into sets by length, e.g., 100–200 µm, for the β-first alloy, and shows that there are more phases for solidification without stirring, with lengths around 0–600 µm and with lengths above 2000 µm, ultimately leading to a smaller average length L_β_ of β. In the alloys where α-Al [34] precipitates first, the histograms showed that the increased number of short β phases precipitating as a result of the forced flow leads to a smaller length L_β_. The histograms confirm the microstructure seen in Figure 5, Figure 6 and Figure 7; closer values of β length for stirring, and fewer very small and fewer very long intermetallics (Figure 5b, Figure 6b and Figure 7b) compared to stirring (Figure 5a, Figure 6a and Figure 7a). For the other alloys, the Si-first, α-Al/Si, β/Si and eutectic point alloys, the length L_β_ was reduced in the range of 10% to 31% and was unchanged for the Si-2-first alloy. The length of β in α-Al/β increases and the number density n_β_ decreases, as is the case with β-first and β-2-first alloys under stirring, but according to the property diagram (Figure 17b) and the precipitation sequence (Table 4), there is a common growth of β and α-Al phases. The explanation of this phenomenon is currently difficult and requires further research, but it seems that perhaps β forming early, through the low mass fraction of α-Al and the excessive amount of liquid, has the potential to grow.

For the α-Al-first and α-Al-2-first alloys the induced fluid flow resulted in an increase (by 130% and 33%) in the number density n_β_ from 6.55 to 15.04 mm^−2^ and from 6.79 to 9.02 mm^−2^, respectively, whereas for the β-first and β-2-first alloys, these values decreased by 71% and 70%. Furthermore, across the specimens, from the center to the outer part (from area A to D), the number density n_β_ changed considerably. The increase in the number density n_β_ was also observed for the Si-first, Si-2-first, α-Al/Si and eutectic point alloys, whereas for α-Al/β and β/Si n_β_, it decreased. An unusual situation can be observed in the β/Si alloy, where both L_β_ and n_β_ decrease under flow. The convergence of results for L_β_ and n_β_ in β-first and β-2-first alloys validates the experiments, methodology, measurements and results obtained in this study.

The directional solidification of Al–Si–Fe alloys [34] by force flow indicated that the shortening of β precipitates has a complex effect on solute segregation, forced convection, intermetallics and dendrite morphology. In [67], an increase in the average length L_β_ and a decrease in the number density n_β_ were found and confirmed in the present study. These seem to be likely due to the rotation of the solid precipitate and a reduction in, or elimination of the constitutional undercooling through the minimization of thermal and solutal diffusion layers at the solid–liquid interface.

### 4.4. Separation of Iron Rich δ-Phases

In the β/Si alloy, δ phases (AlFeSi_T4) were observed, which grew according to the monovariant line (Figure 1, grey dashed line) together with Si crystals as the first precipitating phases (Table 4), both with and without electromagnetic stirring. The average length L_δ_ of δ phases increased under flow by 28%, from 240 µm to 307 µm, and also changed significantly throughout the specimen. The number density n_δ_ decreased significantly, by 58%, from 5.74 mm^−2^ to 2.40 mm^−2^. However, what is remarkable are the changes in the number density n_δ_ throughout the specimen. For solidification without stirring, the number density n_δ_ (Table 3) in area A is 5.36 mm^−2^, in area B 6.05 mm^−2^ and in area C 5.85 mm^−2^, and these values are almost equal (Figure 13a). On the other hand, for electromagnetic stirring n_δ_ is 5.60, 0.09 and 0.70, respectively, so in the outer areas B and C (Figure 13b, Figure 14a and Figure 15), the δ phases are almost absent and have all gathered in the internal part of the specimen (Figure 14a and Figure 15). The microstructural view (Figure 13, Figure 14 and Figure 15) was confirmed by the microstructure parameters (Table 3). In the alloy with the β-Al_5_FeSi phase precipitating as the second one, the number density n_β_ for stirring decreased by 45%, and its value was significant also throughout the specimen, in area A 2.48 mm^−2^, B 11.42 mm^−2^ and C 7.96 mm^−2^. Under a forced flow, the β-Al_5_FeSi phases concentrated in the outer part of the specimen, (area B and C), while the δ phases moved to the central part, area A. The effect of separation discovered between δ, β and Si crystals is completely new.

The δ phases (also denoted Al4, Al_4_FeSi_2_, δ-Al_4_FeSi_2_, Al_3_FeSi_3_, δ(AlFeSi), t(AlFeSi)) are needle- or plate-like shaped, similarly to the β phase, and can therefore be misdiagnosed. However, the δ phases are thicker, have a lower ratio of length to thickness and seem to form more cracks and breaks. According to [68], the δ plates nucleate mainly on the eutectic Si and Al, they grow together with its impingement, branching and deformation, and their formation ends almost simultaneously with the final solidification of the eutectic phases. In the present study, according to the property diagram (Figure 18), δ should nucleate and grow earlier, owing to higher temperature together with the Si crystals, before eutectic Si and Al.

The separation of β and δ, which takes place, seems to provide an opportunity for finding an efficient iron removal method. Li in [69] proposed the use of electromagnetic iron separation in an aluminum melt, Van der Donk and Nijhof [70] proposed mechanical filtration and Matsubara [71] proposed centrifugal separation in which iron phases moved to the edge side of the melt and the central part was purified. Kim and Yoon [72] proposed the elimination of the Fe element in A380 aluminum alloy scrap using electromagnetic force. The present separation of β and δ is completely new and it seems that the application of RMF, complex flows and centrifugal force may play important roles in iron removal, further leading to new technologies.

### 4.5. Reduction of Si Crystals

In the Si-first and Si-2-first alloys, two different areas are clearly visible, the internal area A without Si crystals and the external area B which is rich with Si crystals (Figure 7c,d, Figure 8 and Figure 9). In the Si-first alloy, the melt flow caused a reduction in area B, rich in Si crystals, by changing the boundary from the dimension of about 0.5 of the specimen radius R to 0.8 R, increasing the L_Si_ dimension of the Si crystals (Table 3) by 17% and very strongly reducing (89%) the number density n_Si_. Similarly in the Si-2-first alloy (Figure 9), the boundary between areas A and B changed from 0.7R to 0.85R, the average length of the Si crystals was increased by 20% (Table 3) and very strongly reduced (74%) the number density n_Si_. Electromagnetic stirring definitely modified the growth site of Si crystals and strongly reduced their number in aluminum alloys where Si crystals precipitate first. It is likely that Si is pushed by convection to grow more as eutectics, than as separate Si crystals, which may have a positive effect on alloy properties and castings. In the β/Si alloy, under stirring, the Si crystals grew 51% larger with a 19% reduction in the number density. The largest Si crystals precipitated in the thin outside layer (area C), and crystals also moved to the center (area A) and to the edge of the sample (area C), with the highest number density of 0.630 mm^−2^.

Jie et al. [73] in a cylindrical crucible with a diameter of 60 mm, by bulk solidification, caused the separation of Si crystals in an Al-30 wt.% Si alloy under RMF, where a uniform distribution occurred at 0 mT, while, on stirring, the Si crystals were concentrated to 65–69% Si, in a thin layer of about 8 mm in the evenly solidified sample. The thickness of the separated Si varied from 2 mm to 11 mm for RMF 32 mT and 17 mT, respectively. There was also a slight segregation in the specimen center [73], which was not observed in the present study (Figure 7c,d, Figure 8 and Figure 9). Yu et al. [74] obtained a separation of plate-like Si crystals under electromagnetic stirring in an Al–Si 45 wt.% Si alloy, Ma et al. [75] in a Sn-60 mol% Si (about Sn-74 wt.% Si), Zhu et al. [76] in a Ti-85 wt.% Si alloy and Li Y. et al. [77] obtained the separation of Si crystals in an Al-30 wt.% Si alloy, as a result of mechanical stirring. Ban et al. [78] applied an electromagnetic field to an Al-30 wt.% Si alloy and observed an increase in the average mass and size of primary Si flakes. Zhang et al. [79] in an Al-20.5 wt.% Si alloy applied electric current pulse ECP and observed an increased amount of silicon in the lower part of sample. Li et al. [80] in an Al–28.51 wt.% Si melt, applied a current and decreased the precipitation area of Si. Lv et al. [81], in an Al-45 wt.% Si alloy observed that the electromagnetic field contributed to the positioning of Si crystals in the lower part of the sample. Yoshikawa et al. [82] in Si-55.3 at.% Al and Si-64.6 at.% Al melt with the use of an electromagnetic force observed that Si crystals were separated into the lower half of the specimen. Huang et al. [83] in Sn-30Si alloy, observed the enrichment of primary silicon while removing Al, Fe, Ca, B and P. He et al. [84], through the use of alternating electromagnetic directional solidification (AEM-DS) in an Al-45 wt.% Si alloy, observed the best Si separation by pulling down specimens from furnace. Jiang et al. [85] in electromagnetic directional solidification (EMDS) of Al-35 wt.% Si, Al-45 wt.% Si, Al-55 wt.% Si, and Al-65 wt.% Si alloys observed that a Si-rich region appeared at the bottom of the sample. Xue et al. [86], in an Al-45 wt.% Si alloy, noticed that pulling up from the furnace resulted in Si crystals in the upper part and pulling down resulted in Si crystals in the bottom part. Li, Ren and Fautrelle [87], in Al-18 wt.% Si, found that the high gradient magnetic field separated the Si solute and primary Si at top of the sample. Sun et al. [88] in an Al-30 wt.% Si-10 wt.% Sn melt observed accumulation of primary Si at the bottom and side walls. Zou et al. [89,90] explained, by numerical simulation, that RMF leads to the formation of a swirling flow in the azimuthal direction and then a secondary flow in the meridional plane, playing an important role in the transportation of solute atoms in the axial direction; they also proposed a method for the purification of metallurgical grade Si [91].

The currently observed Si separation in specimens, without stirring, supports findings regarding the important role of the occurring temperature gradient [88,89,90], whereas Si reduction under RMF seems to result from secondary azimuthal flow and the Si crystal movement. The occurrence of a purely eutectic center in the current study agrees with the majority of the before mentioned studies; however, it contradicts [73], where Si crystals were also observed in the center. In the current study, the observed reduction in Si crystals may come from flows reducing the temperature gradient in the melt throughout the sample, and as presented in [88,89,90], the gradient appears to be important. The effects of the temperature gradients, cooling rate and strength of magnetic field require more investigations.

The main effects of the rotating magnetic field RMF on the microstructure were summarized in the Table 5. Depending on their composition, it is possible to predict the modifications to the microstructure of industrial quality alloys that can occur during mold filling and casting solidification as a result of turbulent flow.

### 4.6. Solidification by Stirring

For the α-Al-first and α-Al-2-first alloys, melt stirring slightly reduced the secondary spacing λ_2_ by 6% and 9%, decreased the specific surface of S_v_ by 41% and 32% and increased the eutectic spacing by 20% and 98%, respectively. Flow shortened the β-Al_5_FeSi phases 14% in α-Al-first, whereas in the α-Al-2-first alloy, it stayed unchanged, but the number density n_β_ increased by 130% and 30%. In both alloys, the first precipitating phase is α-Al, and only one for the eutectic reaction. Stirring influenced the growth of α-Al, causing the formation of rosettes (Figure 3b and Figure 4b), minor dendrites and occasionally spheroids, and changed the secondary spacing λ_2_ as well as the specific surface, significantly. According to the ternary phase diagram and property diagram (Figure 1 and Figure 17a), the shortening of β and increase in the number density occur between almost fully grown dendrites, which supports the mechanical interaction between the α-Al dendrites and the moving β. As discussed above, dendrite coarsening occurs during growth, and because of such dense α-Al precipitation, the flow determining the secondary spacing is small, while at the beginning of α-Al growth, there are good conditions to transform dendrites to rosettes. It is also important to mention, that secondary spacing was determined for dendrites which occurred in the stirred alloys, and the fact that they grew means that there is a smaller flow in this region.

For the β-first and β-2-first alloys, melt stirring increased the average length L_β_ of β-Al_5_FeSi 92% and 75%, decreased the number density n_β_ by 71% and 70%, increased the secondary spacing λ_2_ by 60% and 108%, decreased the specific surface S_v_ by 42% and 57% and changed the eutectic spacing. According to thermodynamic calculations (Figure 1 and Figure 17b), the β-Al_5_FeSi phase is the first and only one growing until the eutectic reaction. In the fully liquid alloy, the iron-rich growing phase seems to have the perfect conditions to reach large dimensions, and this tendency is aided by the melt flow, causing a decrease in the number density n_β_ and a strong increase in the average length L_β_. An in-depth analysis of histograms (Figure 16) showed that the flow causes a reduction in the precipitation of very large (Figure 5a and Figure 7a) and especially very small β precipitating in large numbers in the absence of a forced flow. This effect appears to be complicated by the fact that the flow stopped the formation of very long needles and reduced the number density of the very small phases, leading to an equalization of β, but with generally longer needles. The β platelets formed do not appear to stop the flow which modified α-Al and eutectics.

For the Si-first and Si-2-first alloys, melt stirring increased the average length L_Si_ of Si crystals by 17% and 20%, strongly decreased the number density n_β_ by 89% and 74%, increased the secondary spacing λ_2_ by 240% and 42% as well as decreasing the specific surface S_v_ by 30% and 10%, and changing the eutectic spacing. According to thermodynamic calculations (Figure 1, Table 4), the Si crystal is the first and only growing phase until the eutectic reaction. In the fully liquid alloy, the Si crystal growing first appears to have the perfect conditions to reach large dimensions, but this tendency is limited by the melt flow, causing a strong decrease in the number density n_β_ and a slight increase in the average length L_β_. Electromagnetic stirring leads to a significant reduction in the number of Si crystals present and a definite reduction in their growth area, pushing Si to the edges of the samples (Figure 7c,d and Figure 9a,b).

For the α-Al/β alloy, melt stirring increased the secondary spacing λ_2_ by 11% from 71 µm to 79 µm, did not change the specific surface S_v_, increased the average length L_β_ of β-Al_5_FeSi by 17%, decreased the number density n_β_ by 26% and changed the eutectic spacing. According to thermodynamic calculations (Figure 1 and Figure 18), α-Al and β-Al_5_FeSi phases grow together along the monovariant line until the eutectic reaction. The increase in the β length and decrease in the number density, in accordance with current results for α-Al-first and β-first alloys, suggest that during solidification there is enough space between the forming α-Al for enlargement of β under flow. However, with the rapidly increasing amount of α-Al and much smaller amount of β, it is possible that there is a mechanical interaction of both phases and a similarly occurring shortening of phases. Microstructures (Figure 10 and Figure 11) and the phase diagram suggest a very close, connected precipitation of α-Al and β, and show similar shapes for natural and forced convection. The use of stirring may also help form fully shaped dendrites (Figure 11a) specifically as the “bunch of grapes” shaped α-Al. The common growth of α-Al and β strongly determines the whole structure and the specific surface stays unchanged under flow.

For the α-Al/Si alloy, melt stirring increased the secondary spacing λ_2_ by 35%, decreased the specific surface S_v_ by 15%, decreased the average length L_β_ by 10%, increased the number density n_β_ by 110% and changed the eutectic spacing. According to thermodynamic calculations (Figure 1), α-Al and Si phases grow together along the monovariant line until the eutectic reaction. The changes in the alloy under stirring confirm the modification for the previously described alloys. Noteworthy is the modification of large dendrites (Figure 12a) to small rosettes (Figure 12b).

For the β/Si alloy, melt stirring increased the secondary spacing λ_2_ by 255%, decreased the specific surface S_v_ by 53%, decreased the average length L_β_ by 22%, decreased the number density n_β_ by 45%, increased the average length L_δ_ of δ phases by 28%, decreased the number density n_δ_ by 58%, increased L_Si_ of Si crystals by 51%, decreased the number density n_Si_ by 19%, and changed the eutectic spacing. According to thermodynamic calculations (Figure 1, Table 4), δ phases and Si crystals grow together along the monovariant line from liquidus temperature of 610 °C to 596.14 °C, and β-Al_5_FeSi and Si crystals grow together from 596.14 °C till the eutectic reaction at 575 °C. This growth results in the mass fraction of f_δ_ = 1.09 and f_Si_ = 0.97 (Table 4), and in this way, the phases grew in an almost fully liquid alloy. What is new is that both the δ phases and Si crystals grew larger with a lower number density under stirring, and this behavior agrees well with the fact that β-Al_5_FeSi grows as the first and only phase in the β-first alloy and as the first Si crystals in the Si-first alloy. What is completely new is the separation of δ in the center of the specimen (Figure 13b, Figure 14 and Figure 15), the location of Si crystals more in the area near the edge of the sample and the absence of β-Al_5_FeSi in the center of the specimen. It seems that stirring affects both iron-rich phases differently, may separate δ and β in different parts of the cylindrical sample and may also move the Si crystals.

For the eutectic point alloy, melt stirring decreased the secondary spacing λ_2_ by 25%, increased the specific surface S_v_ by 24%, decreased the average length L_β_ by 31%, the increased number density n_β_ by 107% and did not change the eutectic spacing. According to thermodynamic calculations (Figure 1, Table 4), all phases grow at the eutectic reaction. α-Al seems not to follow the pattern known for α-Al rich alloys, and β is shorter in growth between other occurring phases.

Forced convection reduced the secondary dendrite arm spacing λ_2_ in alloys where α-Al is the first phase to grow. In the other alloys, where α-Al may precipitate in or near the eutectic reaction and in small amounts, the dendrite spacing increased. In the eutectic point alloy dendrites should not precipitate, but any that did occur presented reduced spacing at increased specific surface S_v_.

The α-Al phase grew as rosettes with reduced S_v_ in almost all of the alloys studied. Only for α-Al/β, where both α-Al/β grew, did S_v_ not change under stirring, and an atypical structure was observed. The specific surface S_v_ very clearly signals the presence of melt flow in almost all alloys and its effect on the microstructure, which makes it a unique and valid parameter.

The present study confirmed the results of [34] for directional solidification, where forced convection reduced the average length L_β_ of the β phases and increased the number density n_β_ in the alloys where α-Al precipitates first. The results for equiaxed growth [30,67] in experiments performed using the same method and parameters are similar to those presented in Table 1, Table 2 and Table 3. In [67], an increase in β length and decrease in n_β_ were found and these were confirmed in the β-first and β-2-first alloys currently studied. The modification of β phases platelets in directional solidification was explained as partial phase re-melting and intensive nucleation of new intermetallics. The present study and [30,67] suggest an increase in the length of β needles as an effect of a change in solute distribution under flow, whereas the shortening of iron-rich platelets is an effect of mechanical fragmentation by the presence of α-Al or other solid phases. In the presence of forced convection, the elimination of constitutional undercooling through a reduction in solutal and thermal diffusion layers at the liquid–solid interface requires detailed investigation. This explanation seems to fit similarly for the increased length of iron-rich δ phases.

The separation of δ, β and Si crystals shows that the δ phases located in the specimen center grow first, solidifying even through bulk solidifications as the last ones. Si crystals precipitated throughout the liquidus–solidus range mainly at the edge of the sample and in its center, i.e., areas which should solidify at different times. In contrast, β phases formed a ring together with Al–Si eutectics. The sequence of precipitation of the phases present and their distribution throughout the sample suggest a separation due to mechanical interaction and electromagnetic stirring according to a centrifugal force from a horizontal flow, as primary flow generated by the coils, followed by secondary azimuthal flow along the cylindrical sample.

The reduction in Si crystals seems to be caused by the flow lowering the temperature gradient throughout the sample and the mixing and homogenization of the Si concentration, as the Si element moves into the Al–Si eutectics, which are distributed throughout the whole sample.

## 5. Conclusions

The forced flow induced by electric coils produced mainly rosettes instead of equiaxed dendrites, changed the solidification time and secondary dendrite arm spacing λ_2_, decreased the specific surface S_v_ of α-Al and modified Al–Si eutectics;Forced convection caused a decrease in the number density and an increase in the length of β-Al_5_FeSi in alloys, where β iron-rich phases precipitate first, moving initially as only one in the liquid alloy. The free growth of β-Al_5_FeSi in the flowing melt led to longer needles and more even lengths, whereas without stirring, very short and very long plates were observed. The melt flow effect on β-Al_5_FeSi phases depends on the similarly occurring phases and the precipitation sequence;Electromagnetic stirring caused a decrease in the number density and an increase in the length of iron-rich δ-AlFeSi_T4 phases and changed their position across the cylindrical sample in the β/Si alloy, where δ together with Si, is the first precipitating phase;Separation of iron-rich δ and β phases and Si crystals was observed, together with a modification of their dimensions and number density. In the β/Si alloy, in which the iron-rich phases precipitate similarly to the Si crystals along the monovariant line, different specific locations of phases across the cylindrical sample were found and this suggests mechanical separation by force from the horizontal vertical flows generated by electromagnetic stirring;A significant reduction in Si crystals was observed in the alloys where Si crystals precipitate as the first phase. Stirring increased the average length (by 17% and 20%) of Si crystals, but greatly reduced the number density (by 89% and 74%) and moved Si phases into the thin layer outside the cylindrical sample. The reduction in the number of Si crystals appears to be caused by the flow leading to a lower temperature gradient, the mixing and homogenization of the Si concentration and the movement of the Si element to Al–Si eutectics;The present separation of δ and β phases and Si crystals, and their modification are completely new and require further investigation. It seems that the application of RMF as well as complex flows and centrifugal force may play an important role in the removal of iron from foundry aluminum alloys. It may also open new design and production concepts for products with gradient structure and diverse properties, and control of Si content in the production of metallurgical grade silicon for the solar photovoltaic industry;Forced convection changes microstructures in different ways and its efficiency depends on the chemical composition of the alloys, phase growth sequences and the phases present;The present study provides an understanding of the influence of flow on individual phases and gives a picture of what changes can occur in technical alloys with complex chemical compositions.

## Figures and Tables

**Figure 1 materials-16-03304-f001:**
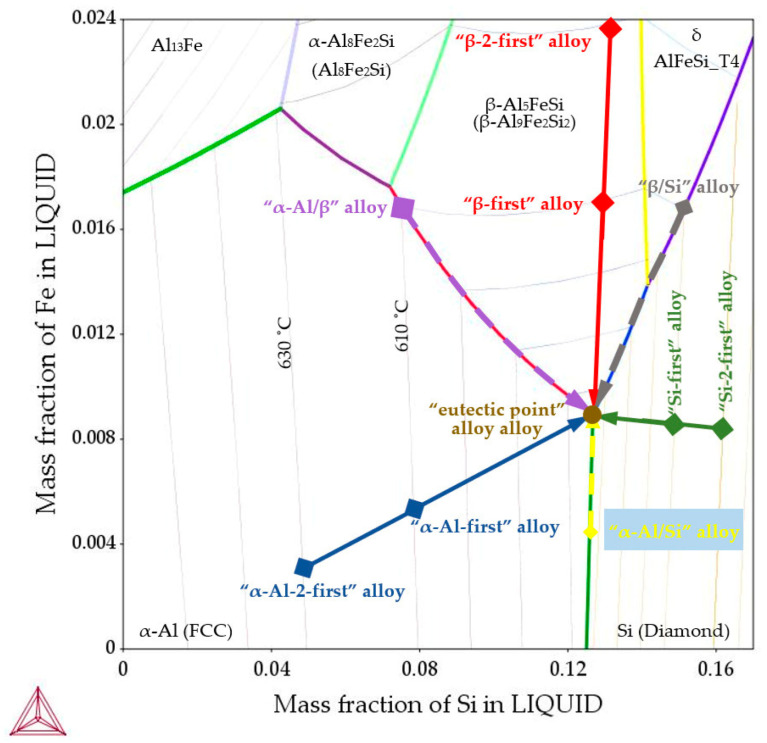
Ternary phase diagram—Al–Si–Fe system. Liquidus projection with the marked solidification paths (Scheil–Gulliver solidification) for studied alloys, e.g., blue continuous line for α-Al-first (Al–Si7.837Fe0.521) and α-Al-2-first (Al–Si4.861Fe0.306) alloys.

**Figure 2 materials-16-03304-f002:**
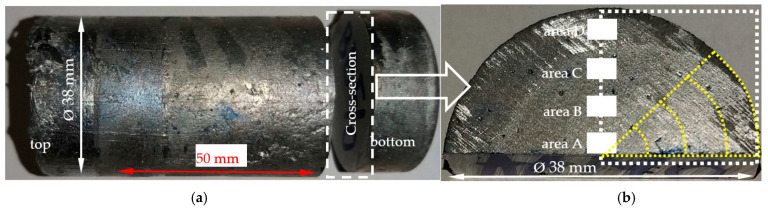
Sample specimen: (**a**) scheme of cutting the cross-section, (**b**) the placement of the microstructure parameters measurement. The white fulfilled rectangles show four areas for the parameter measurement (magnification 50× and 200×). The dotted yellow line shows four arched areas for measurent of large iron intermetallics. The dotted white line rectangle show the measurement area magnified by 50× for other measurements and microstructure presentation.

**Figure 3 materials-16-03304-f003:**
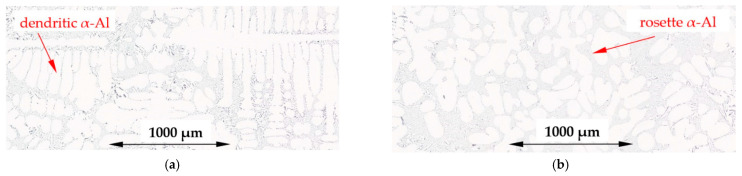
Microstructures of the α-Al-first (Al–Si7.837Fe0.521) alloy specimen solidified: (**a**) without and (**b**) with electromagnetic stirring. LOM, magnification 100×.

**Figure 4 materials-16-03304-f004:**
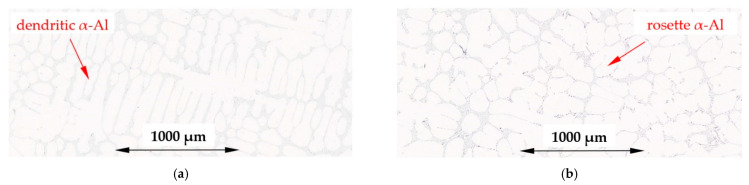
Microstructures of the α-Al-2-first (Al–Si4.861Fe0.306) alloy specimen solidified: (**a**) without and (**b**) with electromagnetic stirring. LOM, magnification 100×.

**Figure 5 materials-16-03304-f005:**
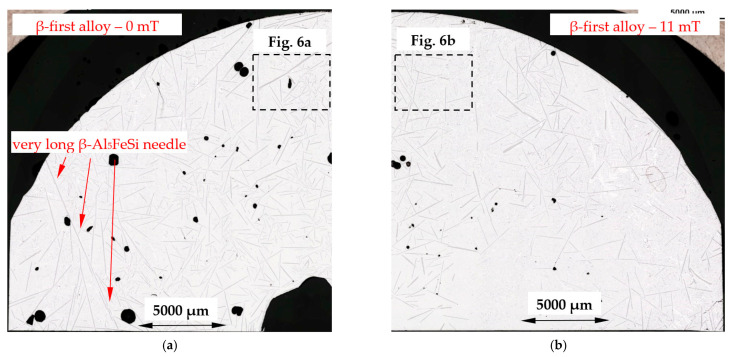
Microstructures of the β-first (Al–Si12.795Fe1.705) alloy specimen solidified: (**a**) without and (**b**) with electromagnetic stirring. LOM, magnification 50×.

**Figure 6 materials-16-03304-f006:**
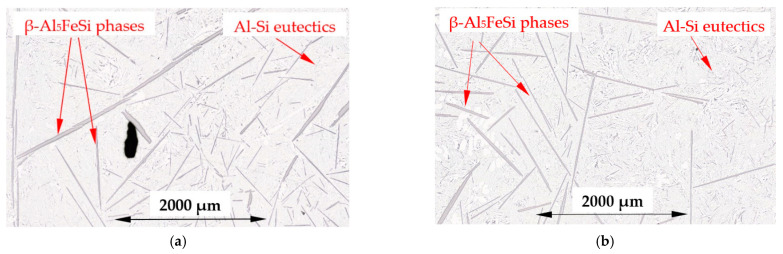
Microstructures of the β-first (Al–Si12.795Fe1.705) alloy specimen solidified: (**a**) without and (**b**) with electromagnetic stirring. LOM, magnification 100×.

**Figure 7 materials-16-03304-f007:**
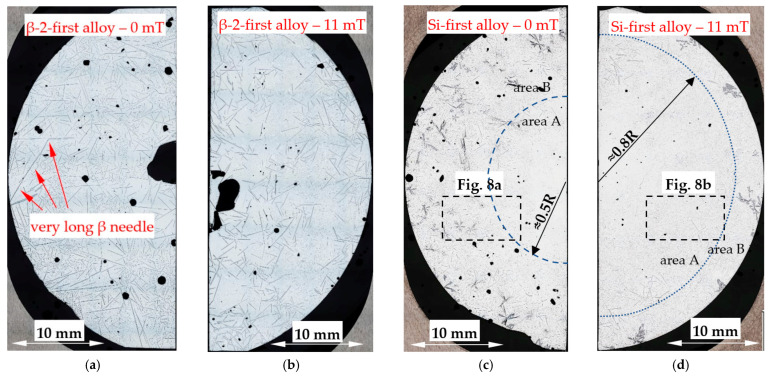
Microstructures of the β-2-first (Al–Si12.911Fe2.372) alloy specimen solidified: (**a**) without, (**b**) with electromagnetic stirring. Microstructures of the Si-first (Al–Si14.877Fe0.871) alloy specimen solidified: (**c**) without, (**d**) with electromagnetic stirring LOM, magnification 25×.

**Figure 8 materials-16-03304-f008:**
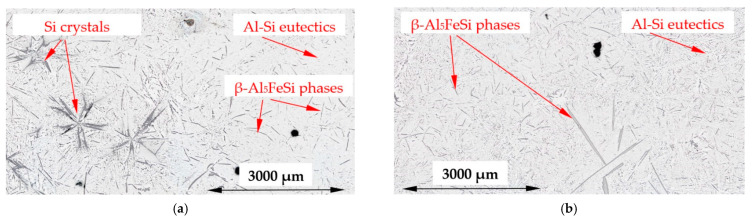
Microstructures of the Si-first (Al–Si14.877Fe0.871) alloy specimen solidified: (**a**) without and (**b**) with electromagnetic stirring. LOM, magnification 100×.

**Figure 9 materials-16-03304-f009:**
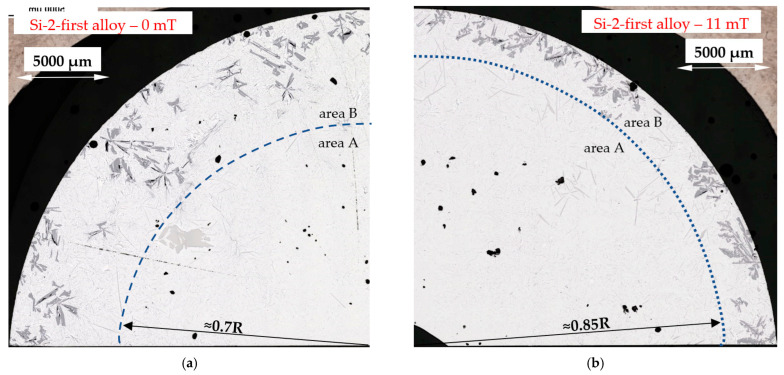
Microstructures of the Si-2-first (Al–Si16.187Fe0.858) alloy specimen solidified: (**a**) without and (**b**) with electromagnetic stirring. LOM, magnification 50×.

**Figure 10 materials-16-03304-f010:**
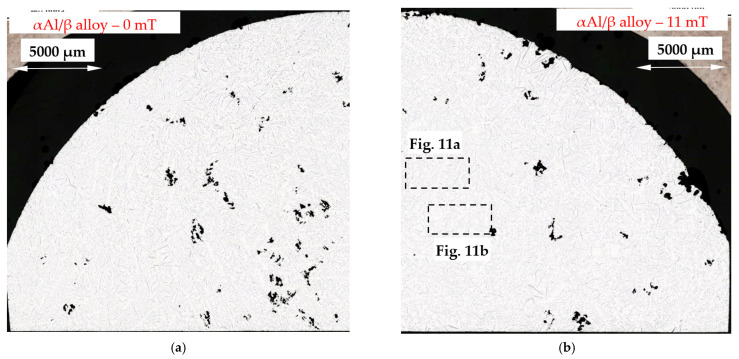
Microstructures of the α-Al/β (Al–Si7.508Fe1.687) alloy specimen solidified: (**a**) without and (**b**) with electromagnetic stirring. LOM, magnification 50×.

**Figure 11 materials-16-03304-f011:**
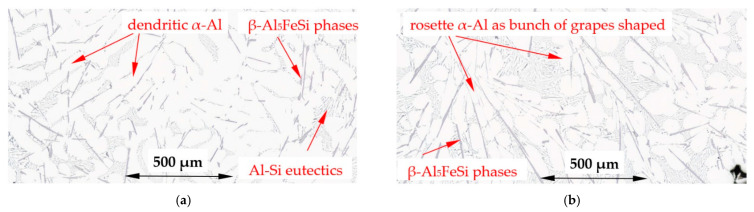
Microstructures of the α-Al/β (Al–Si7.508Fe1.687) alloy specimen solidified by forced convection: (**a**) with dendritic α-Al and (**b**) with rosette α-Al (bunch of grapes shaped). LOM, magnification 100×.

**Figure 12 materials-16-03304-f012:**
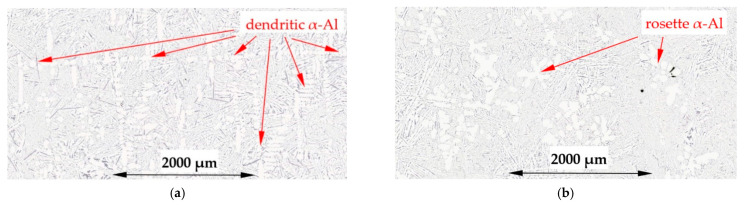
Microstructures of the α-Al/Si (Al–Si12.587Fe0.443) alloy specimen solidified: (**a**) without and (**b**) with electromagnetic stirring. LOM, magnification 100×.

**Figure 13 materials-16-03304-f013:**
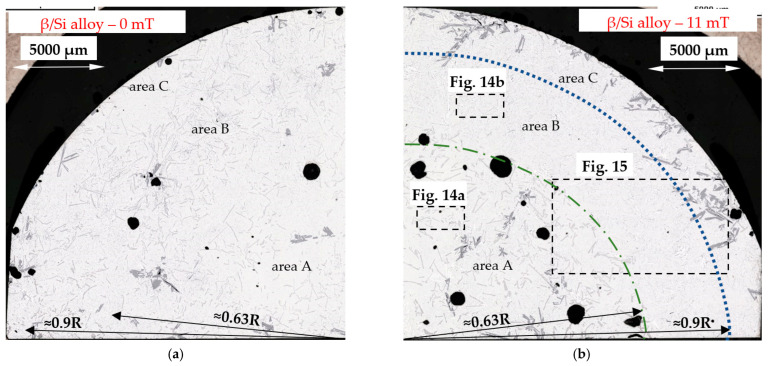
Microstructures of the β/Si (Al–Si15.136Fe1.678) alloy specimen solidified: (**a**) without and (**b**) with electromagnetic stirring. LOM, magnification 50×. The green and the blue lines show the border between areas A, B and C, formed as ab effect of phases separation.

**Figure 14 materials-16-03304-f014:**
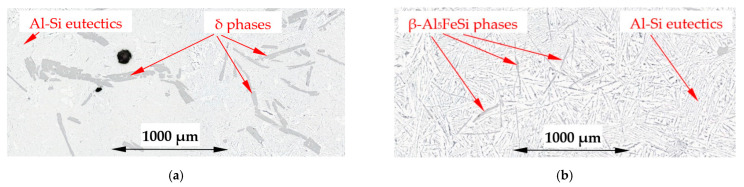
Microstructures of the β/Si (Al–Si15.136Fe1.678) alloy specimen solidified with electromagnetic stirring in: (**a**) area A and in (**b**) area B (details from Figure 13). LOM, magnification 100×.

**Figure 15 materials-16-03304-f015:**
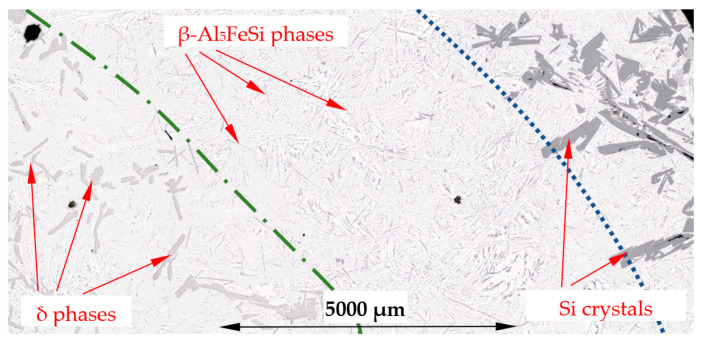
Microstructures of the β/Si (Al–Si15.136Fe1.678) alloy specimen solidified with electromagnetic stirring with well visible areas rich on various separated phases. Detail from Figure 13. LOM, magnification 100×. The green and the blue lines show the border between areas A, B and C, formed as an effect of phases separation.

**Figure 16 materials-16-03304-f016:**
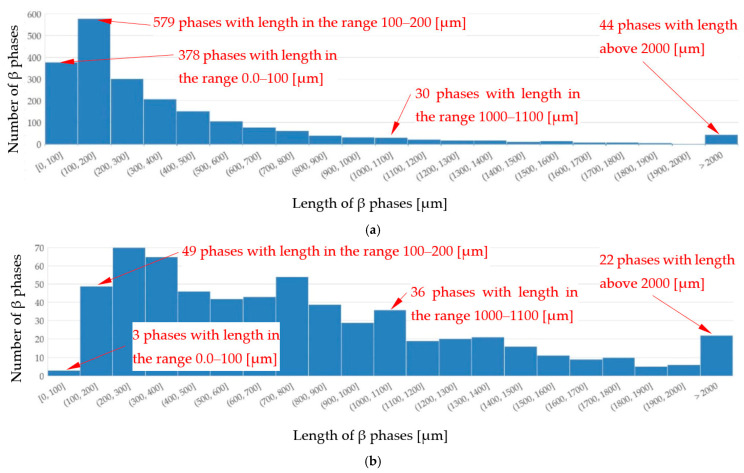
Histogram of measured β-Al_5_FeSi phases in β-first alloy specimens solidified: (**a**) without (0 mT) and (**b**) with electromagnetic stirring (11 mT). Number of phases measured, counted and collected on specified A, B, C and D measurement areas.

**Figure 17 materials-16-03304-f017:**
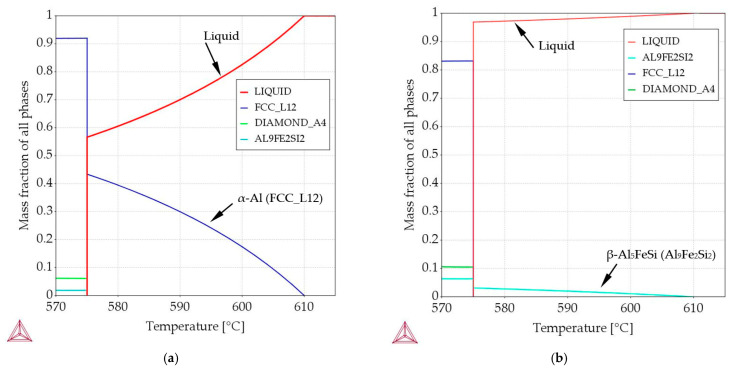
Property diagram for the: (**a**) “α-Al first” alloy (Al–Si7.837Fe0.521) and (**b**) the “β-first” alloy (Al–Si12.795Fe1.705).

**Figure 18 materials-16-03304-f018:**
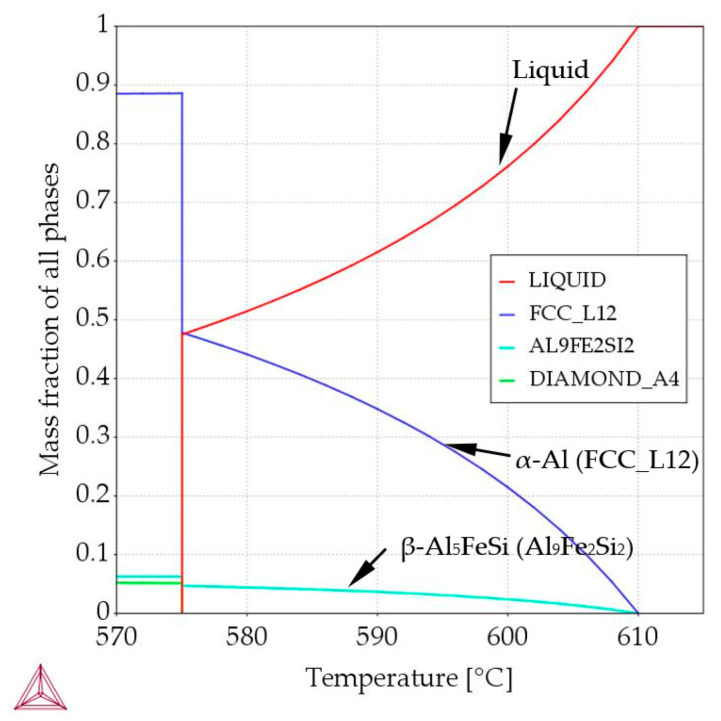
Property diagram for the “α-Al/β” alloy (Al–Si7.508Fe1.687).

**Table 1 materials-16-03304-t001:** Dendrites parameters measured on investigated micrographs of Al–Si–Fe alloys.

Aluminum Alloys	RMF [mT] {Solid. Time [s]}	Dendrites	
		λ_2_ [µm]	S_v_ [µm^−1^]
“**α-Al-first**” alloy Al–Si7.837Fe0.521	0{676}	95 [8.9] (34/332) 93:100:95:91 [9.7:8.9:9.3:8.4]	0.029 [0.003] 0.027:0.032:0.031:0.029 [0.004:0.004:0.002:0.004]
	11{815}	89 [7.5] (37/222) (−6%) 92:94:87:84 [7.0:9.4:5.6:6.1]	0.017 [0.001] (−41%) 0.018:0.016:0.016:0.019 [0.001:0.001:0.001:0.001]
“**α-Al-2-first**” alloy Al–Si4.861Fe0.306	0{795}	112 [10] (38/359) 122:115:120:101 [8.5:12.3:7.6:8.0]	0.022 [0.002] 0.020:0.022:0.021:0.029 [0.001:0.001:0.001:0.002]
	11{779}	102 [7.5] (36/216) (−9%) 104:100:100:105 [8.4:6.4:9.2:6.8]	0.015 [0.001] (−32%) 0.016:0.014:0.015:0.016 [0.001:0.001:0.001:0.001]
“**β-first**” alloy Al–Si12.795Fe1.705	0{519}	22.5 [6.6] (19/194) 17:19:22:51 [2.1:2.4:3.8:3.1]	0.069 [0.027] 0.161:0.131:0.071:0.040 [0.013:0.017:0.007:0.002]
	11{504}	36 [12] (21/220) (60%) 15:32:66:42 [1.8:3.5:7.3:10.3]	0.040 [0.036] (−42%) 0.179:0.076:0.037:0.033 [0.017:0.006:0.009:0.001]
“**β-2-first**” alloy Al–Si12.911Fe2.372	0{525}	22 [9.0] (13/108) 20:-:70:18 [0.1:-:5.2:2.0]	0.081 [0.023] 0.155:0.064:0.056:0.129 [0.007:0.014:0.003:0.013]
	11{469}	60 [12] (27/173) (170%) -:82:68:53 [-:8.8:11.5:12.1]	0.035 [0.006] (−57%) -:0.031:0.035:0.043 [-:0.002:0.004:0.008]
“**Si-first**” alloy Al–Si14.877Fe0.871	0{560}	12 [2.1] (3/27) 12:-:-:- [2.1:-:-:-]	0.087 [0.008] 0.087:-:-:- [0.008:-:-:-]
	11{624}	41 [10.3] (5/59) (240%) 32:54:-:36 [7.2:17.1:-:0.0]	0.061 [0.009] (−30%) 0.089:0.054:-:0.058 [0.001:0.006:-:0.006]
“**Si-2 first**” alloy Al–Si16.187Fe0.858	0{535}	19 [1.2] (2/13) 19:-:-:- [1.2:-:-:-]	0.092 [0.002] 0.092:-:-:- [0.002:-:-:-]
	11{599}	27 [3.7] (4/47) (42%) 24:38:-:- [0.9:0.0:-:-]	0.083 [0.004] (−10%) 0.083:-:-:- [0.004:-:-:-]
“**α-Al/β**” alloy Al–Si7.508Fe1.687	0{841}	71 [4.5] (35/355) 72:71:74:66 [2.5:4.4:4.9:4.9]	0.031 [0.002] 0.030:0.030:0.032:0.030 [0.002:0.002:0.001:0.001]
	11{854}	79 [7.7] (25/252) (11%) 79:72:87:84 [8.6:7.2:4.3:9.7]	0.030 [0.002] (−3%) 0.033:0.030:0.031:0.028 [0.001:0.002:0.002:0.001]
“**α-Al/Si**” alloy Al–Si12.587Fe0.443	0{523}	57 [11.1] (19/186) 85:-:77:45 [10.0:-:9.7:6.2]	0.033 [0.003] 0.034:-:0.034:0.033 [0.002:-:0.003:0.006]
	11{481}	77 [8.3] (31/219) (35%) 61:71:85:80 [12.2:7.4:4.8:6.9]	0.028 [0.003] (−15%) 0.033:0.026:0.029:0.028 [0.005:0.001:0.002:0.001
“**β/Si**” alloy Al–Si15.136Fe1.678	0{556}	20 [3.1] (2/23) 26:17:-:- [0.1:0.1:-:-]	0.076 [0.012] 0.076:-:-:- [0.012:-:-:-]
	11{557}	71 [9.9] (6/50) (255%) 61:-:-:90 [9.9:-:-:5.7]	0.036 [0.006] (−53%) 0.040:-:-:0.030 [0.006:-:-:0.001]
“**eutectic point**” alloy Al–Si12.674Fe0.895	0{471}	69 [9.5] (21/137) 56:72:58:79 [15.3:8.2:11.9:5.8]	0.033 [0.005] 0.040:0.032:0.038:0.029 [0.006:0.002:0.007:0.002]
	11{517}	52 [18.5] (15/140) (−25%) 81:122:84:30 [16.5:0.0:9.2:7.6]	0.041 [0.013] (24%) 0.031:0.037:0.038:0.069 [0.002:0.004:0.002:0.013]

(1) Curly brackets {…} present the solidification time (s); (2) Brackets […] present the standard deviation, (3) Parentheses (…/…) present numbers of grains inspected and dendrite arms counted; (4) Parentheses (…%) present the variation of the parameters in percent under electromagnetic stirring; (5) Dash - presents lack of data caused by absence of, e.g., dendrites or absence of phases; (6) Parameters separated by colon: represent values measured in areas in order A, B, C and D as on Figure 2, if less than four parameters then less areas with phases found and measured (e.g., only two parameters for A and B as on Figure 7 and Figure 9).

**Table 2 materials-16-03304-t002:** Intermetallics and eutectics parameters measured on investigated micrographs of Al–Si–Fe alloys.

Aluminum Alloys	RMF [mT] {Solid. Time [s]}	Fe Phases β-Al_5_FeSi		Al–Si Eutectics
		L_β_ [μm]	n_β_ [mm^−2^]	λ_E_ [μm]
“**α-Al-first**” alloy Al–Si7.837Fe0.521	0{676}	9.40 [0.527] (85) 7.10:8.65:8.49:10.49 [0.249:0.330:0.382:0.630]	6.55 3.70:2.77:5.55:14.19	6.5 [0.43] 5.3:5.3:6.3:8.9
	11{815}	8.07 [0.373] (195) (−14%) 10.05:7.51:6.84:7.65 [0.448:0.265:0.310:0.343]	15.04 (130%) 16.35:11.72:15.11:16.96	7.8 [0.81] (20%) 12.3:6.7:5.7:8.1
“**α-Al-2-first**” alloy Al–Si4.861Fe0.306	0{795}	4.52 [0.224] (88) 4.39:4.19:3.46:5.91 [0.158:0.129:0.149:0.293]	6.79 6.17:3.70:8.95:8.33	2.99 [0.27] 2.82:2.17:2.72:4.31
	11{779}	4.61 [0.176] (117) (2%) 5.71:3.72:4.56:4.52 [0.237:0.085:0.184:0.135]	9.02 (33%) 6.17:5.55:12.34:12.03	5.91 [0.45] (98%) 7.72:4.41:6.18:5.09
“**β-first**” alloy Al–Si12.795Fe1.705	0{519}	407 [51.9] (2130) 326:366:479:418 [32.3:40.6:51.7:65.6]	7.52 27.56:10.77:6.43:5.42	7.10 [0.77] 3.40:5.18:7.74:11.03
	11{504}	780 [53.5] (614) (92%) 859:783:832:723 [50.7:54.0:55.6:51.8]	2.17 (−71%) 2.85:1.68:2.42:2.11	6.68 [0.74] (−6%) 3.21:5.83:8.80:9.88
“**β-2-first**” alloy Al–Si12.911Fe2.372	0{525}	408 [42.7] (1177) 401:297:443:553 [38.2:26.2:42.7:58.6]	8.31 11.1:17.2:7.90:4.42	6.87 [1.59] 3.2:4.52:10.52:10.89
	11{469}	717 [53.2] (359) (76%) 634:705:681:778 [49.1:50.5:52.4:56.2]	2.54 (−70%) 4.07:2.79:2.62:2.16	8.75 [1.56] (27%) 2.42:4.62:12.85:15.08
“**Si-first**” alloy Al–Si14.877Fe0.871	0{560}	394 [23.4] (327) 326:432 [19.1:24.7]	0.58 0.84:0.50	11.54 [1.68] 5.90:9.92:17.58:15.40
	11{624}	286 [25.6] (728) (−27%) 257:323 [21.0:30.0]	1.29 (122%) 1.11:1.59	10.18 [1.51] (−12%) 4.28:9.41:9.45:17.56
“**Si-2-first**” alloy Al–Si16.187Fe0.858	0{535}	241 [18.7] (655) 196:401 [15.6:20.1]	1.16 1.85:0.50	12.82 [1.61] 7.32:9.68:15.59:20.65
	11{599}	236 [19.2] (959) (−2%) 230:281 [19.7:14.4]	1.69 (46%) 2.06:0.75	12.72 [1.27] (−1%) 9.94:7.76:14.70:18.62
“**α-Al/β**” alloy Al–Si7.508Fe1.687	0{841}	109 [8.4] (1840) 92:117:114:115 [8.4:9.1:7.6:8.1]	105 118:106:94:101	4.44 [0.40] 2.83:3.51:5.11:6.73
	11{854}	127 [11.5] (1347) (17%) 177:120:130:99 [14.3:11.1:11.3:8.5]	77 (−26%) 60:69:76:101	9.11 [1.78] (105%) 8.61:7.69:12.38:8.42
“**α-Al/Si**” alloy Al–Si12.587Fe0.443	0{523}	42 [9.4] (1226) 23:39:77:58 [2.1:3.1:22.9:5.7]	70 109:75:41:54	12.50 [1.89] 5.88:10.23:19.44:17.58
	11{481}	38 [3.0] (2592) (−10%) 36:40:38:39 [2.4:2.6:3.2:3.9]	147 (110%) 169:177:127:116	16.83 [1.47] (35%) 10.80:16.44:21.49:19.6
“**β/Si**” alloy Al–Si15.136Fe1.678	0{556}	128 [7.1] (1857) 130:144:109 [6.7:7.4:6.8:-]	13.1 13.8:9.0:20.7	9.83 [1.48] 7.06:5.40:10.41:19.65
	11{557}	100 [9.6] (1830) (−22%) 138:161:170 [7.0:9.8:11.2]	7.22 (−45%) 2.48:11.42:7.96	10.23 [1.60] (4%) 5.97:6.39:13.15:17.32
“**eutectic point**” alloy Al–Si12.674Fe0.895	0{471}	163 [11.6] (1064) 140:155:156:175 [7.6:9.4:12.8:12.0]	15.04 15.29:17.59:14.45:14.57	14.66 [1.54] 9.86:13.48:20.17:15.94
	11{517}	112 [7.7] (2199) (−31%) 131:93:138:104 [9.2:6.6:10.3:5.4]	31.08 (107%) 30.11:41.13:26.50:30.67	15.02 [1.89] (2%) 14.88:17.37:21.04:7.74

(1) Curly brackets {…} present the solidification time (s); (2) Brackets […] present the standard deviation, (3) Parentheses (…%) present the variation of the parameters in percent under electromagnetic stirring; (4) Parentheses (…) present the numbers of intermetallic phases counted; (5) Dash—presents lack of data caused by absence of, e.g., dendrites or absence of phases; (6) Parameters separated by colon: represent values measured in areas in order A, B, C and D as on Figure 2, if less than four parameters then less areas (e.g., only two parameters for A and B as on Figure 7 and Figure 9).

**Table 3 materials-16-03304-t003:** Intermetallics and Si crystals parameters measured on investigated micrographs of Al–Si–Fe alloys.

Aluminum Alloys	RMF [mT]	Fe-Phases (δ-Phases)		Si Crystals	
		L_δ_ [µm]	n_δ_ [mm^−2^]	L_Si_ [µm]	n_Si_ [mm^−2^]
“**Si-first**” alloy Al–Si14.877Fe0.871	0	-	-	898 [41.5] -:898 [-:41.5]	0.067 0.0:0.090
	11	-	-	1047 [94.5] (17%) -:1047 [-:94.5]	0.007 (−89%) 0.0:0.016
“**Si-2-first**” alloy Al–Si16.187Fe0.858	0	-	-	891 [42.9] -:891 [-:42.9]	0.141 0.0:0.277
	11	-	-	1072 [47.1] (20%) -:1072 [-:47.1]	0.037 (−74%) 0.0:0.135
“**β/Si**” alloy Al–Si15.136Fe1.678	0	240 [15.6] (812) 210:250:276 [12.7:15.8:19.0]	5.74 5.36:6.05:5.85	565 [38.5] 714:528:512 [51.7:30.6:35.3]	0.276 0.161:0.248:0.574
	11	307 [20.6] (679) (28%) 306:217:341 [20.9:9.6:18.2]	2.40 (−58%) 5.60:0.09:0.70	851 [60.6] (51%) 759:379:1008 [38.5:18.2:71.5]	0.223 (−19%) 0.196:0.060:0.630

(1) Brackets […] present the standard deviation, (2) Parentheses (…%) present the variation of the parameters in percent under electromagnetic stirring; (3) Dash - presents lack of data caused by absence of, e.g., dendrites or absence of phases; (4) Parameters separated by colon: represent values measured in areas in order A, B, C and D as on Figure 2, if less than four parameters then less areas (e.g., only three parameters for A, B and C as on Figure 13).

**Table 4 materials-16-03304-t004:** Precipitation sequence in studied alloys of Al–Si–Fe system.

Alloy	Reaction	Temperature Range of Reaction	Mass Fraction of Solid Phases [%] (the Rest is Liquid Alloy) at the Temperature [°C]				
			Temperature	*α*-Al	β-Al_5_FeSi	δ-AlFeSi_T4	Al–Si Eutectics
“**α-Al first**” alloy Al–Si7.837Fe0.521	L → *α*-Al + L	610–575.83	575.83	42.75	0	0	0
	L → α-Al + β-Al_5_FeSi + Si	575.83–575.02	575.02	91.96	1.92	0	6.12
“**α-Al-2 first**” alloy Al–Si4.861Fe0.306	L → *α*-Al + L	630–578.86	578.86	68.09	0	0	0.0
	L → α-Al + β-Al_5_FeSi + Si	578.86–575.02	575.02	95.66	1.12	0	3.22
“**β first**” alloy Al–Si12.795Fe1.705	L → β- β-Al_5_FeSi + L	610–575.21	575.21	0	3.10	0	0.0
	L → *α*-Al + β-Al_5_FeSi + Si	575.21–575.02	575.02	83.17	6.33	0	10.50
“**β-2 first**” alloy Al–Si12.911Fe2.372	L → β-Al_5_FeSi + L	630–575.61	575.61	0	5.64	0	0.0
	L → *α*-Al + β-Al_5_FeSi + Si	575.61–575.02	575.02	80.95	8.80	0	10.25
“**Si first**” alloy Al–Si14.877Fe0.871	L → Si + L	610–575.03	575.03	0.0	0.0	0	2.52
	L → *α*-Al + β-Al_5_FeSi + Si	575.03–575.02	575.02	83.69	3.23	0	13.08
“**Si-2 first**” alloy Al–Si16.187Fe0.858	L → Si + L	630–575.03	575.03	0	0	0	4.02
	L → *α*-Al + β-Al_5_FeSi + Si	575.03–575.02	575.02	82.40	3.18	0	14.42
“**α-Al/β**” alloy Al–Si7.508Fe1.687	L → *α*-Al + β-Al_5_FeSi + L	610.0–575.03	575.03	47.81	4.68	0	0.0
	L → *α*-Al + β-Al_5_FeSi + Si	575.03–575.02	575.02	88.60	6.26	0	5.14
“**α-Al/Si**” alloy Al–Si12.587Fe0.443	L → *α*-Al + Si + L	576.04–575.03	575.03	45.03	0.0	0	5.65
	L → *α*-Al + β-Al_5_FeSi + Si	575.03–575.02	575.02	87.37	1.64	0	10.99
“**β/Si**” alloy Al–Si15.136Fe1.678	L → δ-AlFeSi_T4 + Si + L	610–596.14	596.14	0.0	0	1.09	0.97
	L → β-Al_5_FeSi + Si + L	596.14–575.03	575.03	0.0	3.11	0.0	2.70
	L → *α*-Al + β-Al_5_FeSi + Si	575.03–575.02	575.02	80.87	6.23	0.0	12.90
“**eutectic point**” alloy Al–Si12.674Fe0.895	L → *α*-Al + β-Al_5_FeSi + Si	575.03–575.02	575.02	85.86	3.31	0.0	10.83

**Table 5 materials-16-03304-t005:** Main effect of forced convection on microstructure parameters by electromagnetic stirring RMF in Al–Si–Fe alloys.

Aluminum Alloys (T_L_ [°C])	Main Effect of Forced Convection on Microstructure Parameters by Electromagnetic Stirring RMF	Parameters Change in %						
		λ_2_ [µm]	S_v_ [µm^−1^]	L_β_ [μm]	n_β_ [mm^−2^]	λ_E_ [μm]	L_Si_ [µm]	n_Si_ [mm^−2^]
**α-Al-first** (610 °C)	Decreased specific surface of α-Al dendrites and increased number density of β-phases	−6%	−41%	−14%	130%	20%	-	-
**α-Al-2-first** (630 °C)		−9%	−32%	2%	33%	98%	-	-
**β-first** (610 °C)	Increased length and decreased number density of β-phases	60%	−42%	92%	−71%	−6%	-	-
**β-2-first** (630 °C)		170%	−57%	76%	−70%	27%	-	-
**Si-first** (610 °C)	Decreased number density of Si crystals and placement in the outside of specimen	240%	−30%	−27%	122%	−12%	17%	−89%
**Si-2 first** (630 °C)		42%	−10%	−2%	46%	−1%	20%	−74%
**α-Al/β** (610 °C)	Only slightly changed α-Al dendrites and β-phases	11%	−3%	17%	−26%	105%	-	-
**α-Al/Si** (610 °C)	Increased number density of β-phases	35%	−15%	−10%	110%	35%	-	-
**β/Si** (610 °C)	Separation of δ-phases Increased size of Si crystals by decreased number density	255%	−53%	−22%	−45%	4%	51%	−19%
**eutectic** **point** (576 °C)	Decreased length and increased number density of β-phases	−25%	24%	−31%	107%	2%	-	-

## Data Availability

The data presented in this study are available on request from the corresponding author.

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
