# Peer review of "Distribution and Morphology of α-Al, Si and Fe-Rich Phases in Al–Si–Fe Alloys under an Electromagnetic Field"

_materials, 2023, doi:10.3390/ma16093304_

Round 1

Reviewer 1 Report

The author has done the quantum of work. The obtained results can be understood intuitively the manuscript can be given further consideration with some modifications and additions.

My suggestions are listed below.

1.      A concise and factual abstract is required to publish in materials journal. The abstract should state briefly the purpose of the research, applications, methodology, the principal results and major conclusions.  Please modify.

2.      Figure 1 citation is missing.

3.      Figure 2 scale bar is missing.

4.      In Figure 3,4 and 5 the author labelled alpha Aluminium and different elements. How do you confirm this without EDS analysis?

5.     It is important there be a dedicated limitation paragraph/section that highlights the shortcomings of the experimental, computational and/or overall procedures

Author Response

Dear Reviewer

Thank you very much for the reading, reviewing and preparing remarks and tips, concerning papers subject, study and edition.

The author has analyzed and followed all of the comments. There are answers and proposed modifications in the manuscript in ITALIC.

  1. A concise and factual abstract is required to publish in materials journal. The abstract should state briefly the purpose of the research, applications, methodology, the principal results and major conclusions.  Please modify.

The Abstract has been modified according suggestions.

OLD version

The solidification of AlSiFe alloys was studied using a rotating magnetic field to understand the effect of forced flow through a low cooling rate and low temperature gradient. Using thermodynamic calculations, the composition of alloys was chosen to enable independent growth or joint growth of occurring α-Al, β-Al5FeSi, δ-AlFeSi_T4 phases and Si crystals. Stirring produced mainly rosettes instead of equiaxed dendrites, changed the solidification time and secondary dendrite arm spacing λ2, decreased the specific surface Sv of α-Al and modified the AlSi eutectic spacing. During independent growth, in alloys where β starts to precipitate first, the flow resulted in larger and fewer β with more even lengths. The melt flow caused an increase in the length and a decrease in the number density of iron rich δ-AlFeSi_T4 phases, and gathered them inside the sample of the β/Si alloy, where δ together with Si, were the first precipitating phases. The separation of δ and β phases and Si crystals was found by their joint growth along the monovariant line. A reduction in the amount of Si crystals and the formation of a thin Si-rich layer outside the sample was observed in the hypereutectic alloy. The separation and reduction of phases may play a role in the removal of Fe from AlSi alloys and the control of Si in materials for the solar photovoltaic industry.

NEW version

Natural convection is present in all liquid alloys while forced convection may be applied as the method to improve materials properties. To understand the effect of forced convection, the solidification in simple cylindrical samples was studied using a rotating magnetic field by low cooling rate and low temperature gradient. The composition of AlSiFe alloys was chosen to enable independent growth or joint growth of occurring α-Al, β-Al5FeSi, δ-AlFeSi_T4 phases and Si crystals and analysis of structure modifications. Stirring produced rosettes instead of equiaxed dendrites, changed secondary dendrite arm spacing, specific surface of α-Al and modified β-Al5FeSi. The melt flow caused an modification of iron rich δ-AlFeSi_T4 phases, and gathered them inside the sample of the β/Si alloy, where δ together with Si, were the first precipitating phases. The separation of δ and β phases and Si crystals was found by their joint growth along the monovariant line. A reduction in the amount of Si crystals and the formation of a thin Si-rich layer outside the sample was observed in the hypereutectic alloy. The separation and reduction of iron rich phases may play a role in the removal of Fe from AlSi alloys and the control of Si may be applied in materials for the solar photovoltaic industry

  1. Some Figure 1 citation is missing. 

The citation has been added.

Figure 1. Ternary phase diagram—Al-Si-Fe system [27]. Liquidus projection with the marked solidification paths (Scheil–Gulliver solidification) for studied alloys, e.g. blue continuous line for α-Al-first (AlSi7.837Fe0.521) and α-Al-2-first (AlSi4.861Fe0.306) alloys.

  1. The Figure 2 scale bar is missing. 

There on the Figure 2 were dimension of the specimens presented, for the length and diameter. The scale has been added, in red.

  1. In Figure 3,4 and 5 the author labelled alpha Aluminium and different elements. How do you confirm this without EDS analysis?.

From the methodology, is well know that author has NOT studied industrial quality alloys, not unknown composition, or not and approximated or measured composition of the alloy. The studied alloys were synthetic, prepared and melted from pure, high quality elements. The alloys were melted in new clean graphite crucibles under argon atmosphere, so the alloys composition and structure was perfectly controlled.

On the microstructure are seen only phase/elements which were planed and predicted from phase diagram calculated. It is not alloys with composition Al-Si-X-Y-Z-…, where different unknown precipitate could occurs. The studied aluminum alloys and its structure are well known, so the identification is easily, and especially for the author, who previously studied similar structures.

The problem was described in the paper in line 179-188

In the present paper, the author studied precipitations, such as AlSi eutectics, dendritic α-Al, needle or platelet shaped β-Al5FeSi and δ-AlFeSi_T4 phases; these are all well-known from many other studies (e.g. in [1–6,25,26,28–34]) concerning aluminum alloy phases. The precipitation sequence was first considered theoretically according to the Al-Si-Fe phase diagram, but the exact precipitation of only one first phase (e.g. α-Al in the “α-Al-first” alloy) or both (e.g. dendritic α-Al and β-Al5FeSi in “α-Al/β” alloy) required a precise determination of the alloy composition, which was calculated using the Thermo-Calc software [27]. Widely used by materials scientists and engineers, the software was used to determine property diagrams, the sequence of precipitation of phases, the ternary phase diagram and the Scheil solidification calculations

  1. It is important there be a dedicated limitation paragraph/section that highlights the shortcomings of the experimental, computational and/or overall procedures.

The paper, especially discussion is long, because appropriate results from previous works has to be mentioned for readers and discussed for results interpretation.

The paragraph results was earlier shortened and partially moved into discussion, in order to avoid repetition. The lack of mentioning result might be negatively marked by other Reviewer. The mentioned results are important for clearness and correct interpretation.

The DISCUSSION paragraph has been shortened in:

4.2. Eutectics

4.5 Reduction of Si crystals

And for better understanding of conditions and results, the Table 5 was added,

Table 5. Main effect of forced convection on microstructure parameters by electromagnetic stirring RMF in AlSiFe alloys

Numbers of the paragraphs 4…., were also changed.

The paragraph 3.3 Precipitation sequence might be shortened. But, the author shortened it before submitting. There is no description for precipitation for all alloys, only for three. Of course it might be without description, only with diagrams and table, but it could be difficult to understand  for readers not familiar with phases diagrams.

Best regards

Piotr Mikołajczak

Reviewer 2 Report

1. The manuscript is comprehensive, but the manuscript is too wordy, which is easy to lost and hard to follow. I would suggest the authors to simplify the manuscript and highlight the key findings of the study. It is suggested to use a table to describe the samples with conditions. 

2. Could the author explain how they determine the composition? What is the relationship between calculated and experimental composition? Would the calculated composition represent the experimental sample compositions?

3. What is the implication of current study for the research field?

Author Response

Dear Reviewer

Thank you very much for the reading, reviewing and preparing remarks and tips, concerning papers subject, study and edition.

The author has analyzed and followed all of the comments. There are answers and proposed modifications in the manuscript in ITALIC.

  1. The manuscript is comprehensive, but the manuscript is too wordy, which is easy to lost and hard to follow. I would suggest the authors to simplify the manuscript and highlight the key findings of the study. It is suggested to use a table to describe the samples with conditions. 

The paper, especially discussion is long, because appropriate results from previous works has to be mentioned for readers and discussed for results interpretation.

The paragraph results was earlier shortened and partially moved into discussion, in order to avoid repetition. The lack of mentioning result might be negatively marked by other Reviewer. The mentioned results are important for clearness and correct interpretation.

The DISCUSSION paragraph has been shortened in:

4.2. Eutectics

4.5 Reduction of Si crystals

As suggested by the Reviewer, for better understanding of conditions and results, the Table 5 was added,

Table 5. Main effect of forced convection on microstructure parameters by electromagnetic stirring RMF in AlSiFe alloys

Numbers of the paragraphs 4…., were also changed.

The paragraph 3.3 Precipitation sequence might be shortened. But, the author shortened it before submitting. There is no description for precipitation for all alloys, only for three. Of cource it might be without description, only with diagrams and table, but it could be difficult to understand  for readers not familiar with phases diagrams.

  1. Could the author explain how they determine the composition? What is the relationship between calculated and experimental composition? Would the calculated composition represent the experimental sample compositions?

First, the composition was read from Al-Si-Fe phase diagram (Fig. 1), secondly precise composition was calculated in Thermocalc, Al, Si, Fe content was determined by many iterative calculation, and as third alloys were prepared from pure components. The calculated and the experimental composition are exact the same.

From the methodology, is well know that author has NOT studied industrial quality alloys, not unknown composition, or not and approximated or measured composition of the alloy. The studied alloys were synthetic, prepared and melted from pure, high quality elements. The alloys were melted in new clean graphite crucibles under argon atmosphere, so the alloys composition and structure was perfectly controlled.

On the microstructure are seen only phase/elements which were planed and predicted from phase diagram calculated. It is not alloys with composition Al-Si-X-Y-Z-…, where different unknown precipitate could occurs. The studied aluminum alloys and its structure are well known, so the identification is easily, and especially for the author, who previously studied similar structures.

The problem was described in the paper in line 179-188

In the present paper, the author studied precipitations, such as AlSi eutectics, dendritic α-Al, needle or platelet shaped β-Al5FeSi and δ-AlFeSi_T4 phases; these are all well-known from many other studies (e.g. in [1–6,25,26,28–34]) concerning aluminum alloy phases. The precipitation sequence was first considered theoretically according to the Al-Si-Fe phase diagram, but the exact precipitation of only one first phase (e.g. α-Al in the “α-Al-first” alloy) or both (e.g. dendritic α-Al and β-Al5FeSi in “α-Al/β” alloy) required a precise determination of the alloy composition, which was calculated using the Thermo-Calc software [27]. Widely used by materials scientists and engineers, the software was used to determine property diagrams, the sequence of precipitation of phases, the ternary phase diagram and the Scheil solidification calculations

  1. What is the implication of current study for the research field?

The modification of microstructure may be supposed in industrial quality alloys, depending of its composition, as an effect of turbulent flow during mold filling and casting solidification. Applied in continuous casting crystallizations equipped with electromagnetic stirring offer better quality of casting, and current paper shows the possible modifications for hypo hypereutectic alloys under stirring.

Author has found separation of δ-phases, and its importance comes from known problem with Fe in aluminum alloys. Found out separation may open new way in Fe removal from scrap Al alloys.

  1. Another modifications

The Abstract has been modified.

OLD version

The solidification of AlSiFe alloys was studied using a rotating magnetic field to understand the effect of forced flow through a low cooling rate and low temperature gradient. Using thermodynamic calculations, the composition of alloys was chosen to enable independent growth or joint growth of occurring α-Al, β-Al5FeSi, δ-AlFeSi_T4 phases and Si crystals. Stirring produced mainly rosettes instead of equiaxed dendrites, changed the solidification time and secondary dendrite arm spacing λ2, decreased the specific surface Sv of α-Al and modified the AlSi eutectic spacing. During independent growth, in alloys where β starts to precipitate first, the flow resulted in larger and fewer β with more even lengths. The melt flow caused an increase in the length and a decrease in the number density of iron rich δ-AlFeSi_T4 phases, and gathered them inside the sample of the β/Si alloy, where δ together with Si, were the first precipitating phases. The separation of δ and β phases and Si crystals was found by their joint growth along the monovariant line. A reduction in the amount of Si crystals and the formation of a thin Si-rich layer outside the sample was observed in the hypereutectic alloy. The separation and reduction of phases may play a role in the removal of Fe from AlSi alloys and the control of Si in materials for the solar photovoltaic industry.

NEW version

Natural convection is present in all liquid alloys while forced convection may be applied as the method to improve materials properties. To understand the effect of forced convection, the solidification in simple cylindrical samples was studied using a rotating magnetic field by low cooling rate and low temperature gradient. The composition of AlSiFe alloys was chosen to enable independent growth or joint growth of occurring α-Al, β-Al5FeSi, δ-AlFeSi_T4 phases and Si crystals and analysis of structure modifications. Stirring produced rosettes instead of equiaxed dendrites, changed secondary dendrite arm spacing, specific surface of α-Al and modified β-Al5FeSi. The melt flow caused an modification of iron rich δ-AlFeSi_T4 phases, and gathered them inside the sample of the β/Si alloy, where δ together with Si, were the first precipitating phases. The separation of δ and β phases and Si crystals was found by their joint growth along the monovariant line. A reduction in the amount of Si crystals and the formation of a thin Si-rich layer outside the sample was observed in the hypereutectic alloy. The separation and reduction of iron rich phases may play a role in the removal of Fe from AlSi alloys and the control of Si may be applied in materials for the solar photovoltaic industry

The citation has been added to Figure 1..

Figure 1. Ternary phase diagram—Al-Si-Fe system [27]. Liquidus projection with the marked solidification paths (Scheil–Gulliver solidification) for studied alloys, e.g. blue continuous line for α-Al-first (AlSi7.837Fe0.521) and α-Al-2-first (AlSi4.861Fe0.306) alloys.

There on the Figure 2, were dimensions of the specimens presented, for the length and diameter. The scale bar has been added, in red.

Best regards

Piotr Mikołajczak
